# Size-segregated analysis of PAHs in Urban air: Source apportionment and health risk assessment in an Urban canal-adjacent environment

**Siwatt Pongpiachan** [1,2]*, **Danai Tipmanee**[3], **Chukkapong Khumsup**[4], **Phoosak Hirunyatrakul**[4], **Muhammad Zaffar Hashmi**[5], **Saran Poshyachinda**[1]

**1** National Astronomical Research Institute of Thailand (Public Organization), Chiangmai, Thailand, **2** NIDA Center for Research & Development of Disaster Prevention & Management, School of Social Development and Strategic Management, National Institute of Development Administration (NIDA), Bangkok, Thailand, **3** Faculty of Technology and Environment, Prince of Songkla University, Phuket, Thailand, **4** Bara Scientific Co., Ltd., Bangkok, Thailand, **5** Department of Environmental Health and Management, Health Services Academy, Islamabad, Pakistan

* siwatt.p@nida.ac.th, pongpiajun@gmail.com

## Abstract

This study examines the distribution, origins, and health hazards of polycyclic aromatic hydrocarbons (PAHs) across six particle size fractions obtained from an urban rooftop location in Bangkok, Thailand. We collected PM samples using a six-stage cascade impactor at a canal boat port, trapping PAHs in particle sizes ranging from ultrafine ($PM_{0.65-1.1}$) to coarse ($PM_{7.0 \text{ and beyond}}$) over an 11-week period. We utilized gas chromatography-mass spectrometry to quantify twelve PAH congeners. Results indicated that PAHs primarily concentrate in fine particles ($PM_{2.1-3.3}$), with traffic emissions from gasoline and gasoline cars being the principal sources, augmented by emissions from diesel canal boats and industrial activities. The health risk assessment showed that the lifetime lung cancer risk (LLCR) values for all particle sizes were less than $1 \times 10^{-6}$. This means that PAH exposure in this area has a very low cancer risk. Principal Component Analysis (PCA) and Positive Matrix Factorization (PMF) found traffic and industrial emissions as the primary sources of PAHs, with canal boats accounting for 5% of the total. These findings highlight the necessity of specific emission control regulations and advocate for the implementation of cleaner fuel alternatives and electric propulsion in canal transit to enhance urban air quality in Bangkok.

## 1. Introduction

Urban air pollution has become a significant public health and environmental concern, especially in rapidly developing cities like Bangkok, the capital of Thailand. Bangkok's extensive canal system, known locally as "klongs," has long been an essential part of the city's transportation and logistics network [1,2]. In recent years, as road traffic has increased, so too have air pollution levels, prompting efforts to find alternative, more sustainable modes of transport [3]. Canal boats are often promoted as an effective solution to reduce road traffic and alleviate

**Data availability statement:** All relevant data are within the paper and its Supporting Information files.

**Funding:** This research has received funding support from the NSRF via the Program Management Unit for Human Resources & Institutional Development, Research and Innovation [grant number B11F670110], Ministry of Higher Education, Science, Research and Innovation, Thailand. It is crucial to underline that the funders had no role in study design, data collection and analysis, decision to publish, or preparation of the manuscript.

**Competing interests:** The authors have declared that no competing interests exist.

air pollution challenges, offering a faster and more efficient means of moving goods and people across the city [4].

Despite their potential environmental benefits, most canal boats rely on gasoline or diesel engines, which emit a variety of harmful air pollutants, including nitrogen oxides ($NO_x$), particulate matter (PM), carbon monoxide (CO), and polycyclic aromatic hydrocarbons (PAHs) [5–8]. PAHs are a group of toxic organic chemicals formed during the incomplete combustion of fossil fuels, industrial wastes, and biomass materials [9–11]. These compounds often attach to fine particles, such as $PM_{2.5}$, which can travel long distances through the air and pose serious health risks when inhaled. Exposure to particulate PAHs has been linked to respiratory and cardiovascular diseases, as well as an increased risk of cancer [12–16]. Smaller particles, especially ultrafine particles, can penetrate deeply into the respiratory system, making them particularly harmful [17,18]. Vulnerable populations, such as children and the elderly, are at even greater risk from prolonged exposure to elevated air pollution levels [19–21]. The impact of canal boat emissions is especially concerning in areas where boats operate near residential neighborhoods, markets, and boat terminals. Emissions often accumulate in these areas, exposing passengers and local communities to higher levels of pollution. Inadequately maintained engines or the use of outdated technology in canal boats can exacerbate this problem, contributing to air quality degradation and affecting aquatic ecosystems in the canals [22,23].

Although several studies have examined air pollution from ships and boats, most have focused on bulk emissions without considering the chemical composition of different particle sizes [5–8,24]. The particle size distribution of aerosols plays a crucial role in determining both their health impacts and their potential sources. Fine particles (e.g., $PM_{2.5}$) tend to carry higher concentrations of PAHs, making them more hazardous to human health, while larger particles can still contribute to localized pollution under certain conditions [25–27]. For instance, Wang et al. (2024) reported a bimodal distribution of PAHs in Ningbo, China, with peaks in both fine and coarse particles, and noted the higher toxicity risks of coarse particles for children [28]. Shen et al. (2019) observed that fine particles (<1.1 μm) carried the majority of PAHs in China, with elevated cancer risks linked to industrial emissions in winter [29]. In Delhi, Gupta et al. (2011) identified that PAHs were primarily concentrated in the smallest particles (<0.7 μm), with vehicular emissions and biomass burning as major sources [30]. Similarly, Pham et al. (2024) found that biomass burning in Hanoi significantly increased PAH concentrations, with smaller particles (1.34–1.5 μm) dominating during rice straw burning events [31]. Cao et al. (2020) demonstrated that finer particles (0.7–2.1 μm) carried the highest PAH concentrations in Northern China, posing greater cancer risks for children, particularly through inhalation exposure [32].

These findings underscore the importance of understanding the size-specific distribution of PAHs in urban environments, as this knowledge is essential for identifying pollution sources and developing effective mitigation strategies. Therefore, this study aims to (*i*) chemically characterize PAHs in six particle size categories ($PM_{0.65–1.1}$, $PM_{1.1–2.1}$, $PM_{2.1–3.3}$, $PM_{3.3–4.7}$, $PM_{4.7–7.0}$, and $PM_{7.0 \text{ and above}}$) collected at an air quality observatory site adjacent to canal boat terminals, (*ii*) identify the potential sources of PAHs using Principal Component Analysis (PCA) coupled with Positive Matrix Factorization (PMF), and (*iii*) assess the health risks associated with exposure to these carcinogenic substances.

## 2. Materials and methods

### 2.1 Sampling site

The National Institute of Development Administration (NIDA) Air Quality Observatory Site (NAQOS) is conveniently positioned on the NIDA campus in Bangkok, Thailand (see Fig 1). The campus has a mixture of urban and suburban areas, making it suitable for monitoring

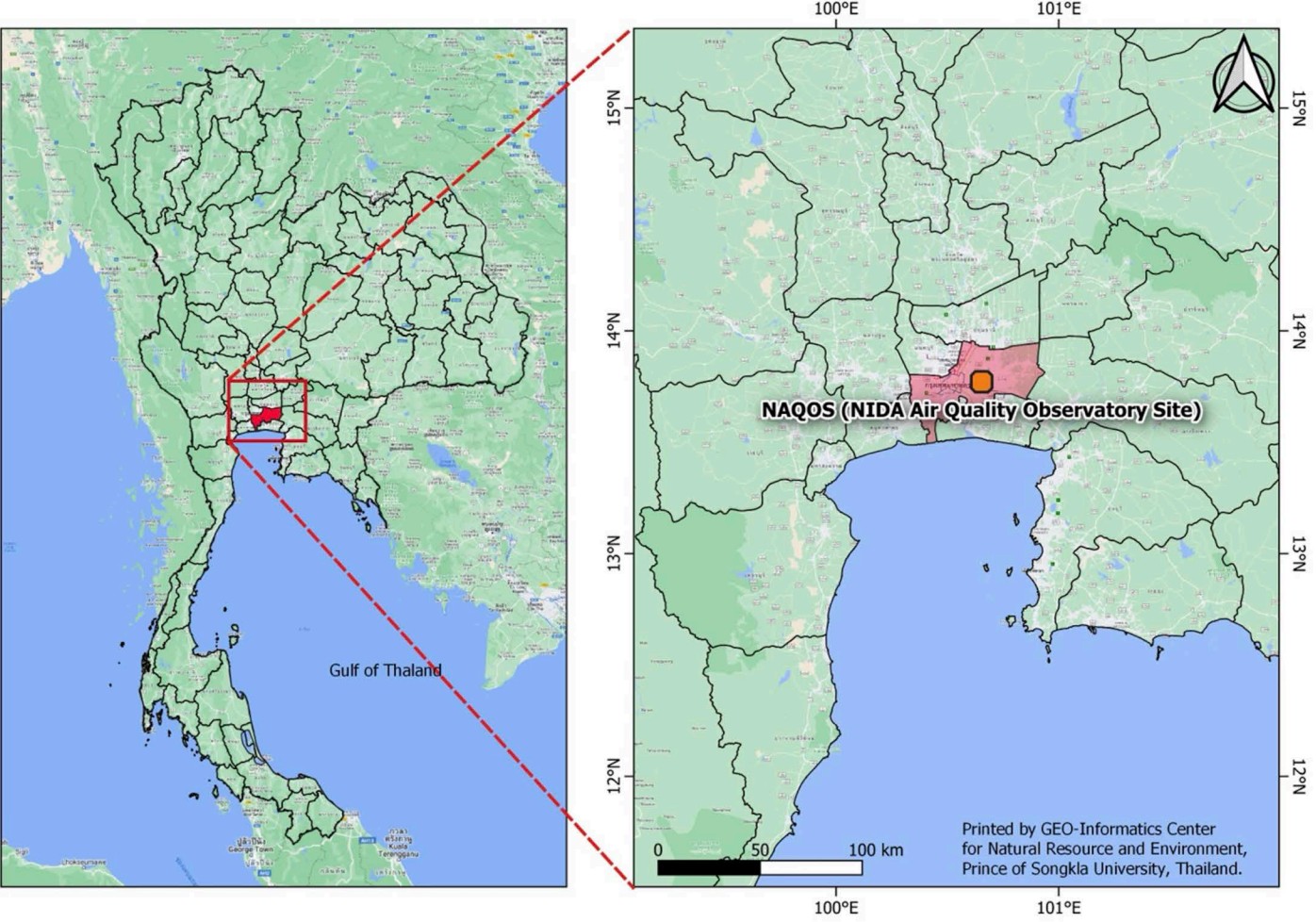

**Fig 1. A map of NIDA air quality observatory site (NAQOS) in Bangkok, Thailand.**

air quality from a variety of potential sources. The NAQOS is positioned on the rooftop of the Navamindradhiraj Building at an approximate height of 60 m. This observatory site was established to continuously monitor and assess air quality in the Bangkok metropolitan area, which is notorious for air quality issues owing to urbanisation, traffic congestion, and industrial operations. The information gathered at this site is used for environmental studies and policymaking targeted at improving air quality and public health.

PM samples across six size fractions (i.e., $PM_{0.65-1.1}$, $PM_{1.1-2.1}$, $PM_{2.1-3.3}$, $PM_{3.3-4.7}$, $PM_{4.7-7.0}$, and $PM_{7.0\ and\ above}$) were collected using a six-stage cascade impactor setup from January 2nd to April 7th, 2023. Sampling took place at the NAQOS, which is equipped to capture ambient air quality data under standard environmental conditions. A total of 66 air samples (i.e., 6×11) were collected continuously from 9:00 AM Monday to 9:00 AM Saturday over an 11-week period, amounting to a total sampling duration of 120 hours per week, to ensure consistency and capture potential variations in particulate composition across weekdays. The sampling frequency and time frame were chosen to provide a comprehensive dataset of PM distribution and PAH concentration in relation to environmental and anthropogenic factors. Each PM fraction was collected onto pre-conditioned filters at designated flow rates to allow subsequent chemical analysis of particulate-bound compounds. In addition, it is also crucial to

underline that all meteorological parameters were obtained from the weather station adjacent to NAQOS, which has been operated by Thai Meteorological Department Automatic Weather System (http://www.aws-observation.tmd.go.th/rprt/weatherDays).

## 2.2 Sampling protocol

To accumulate airborne aerosols with different particle sizes, Tisch Environmental employed ambient viable Andersen cascade impactors. The cascade impactor (TE-10–800) provided six-stage configurations (i.e., $PM_{0.65-1.1}$, $PM_{1.1-2.1}$, $PM_{2.1-3.3}$, $PM_{3.3-4.7}$, $PM_{4.7-7.0}$, and $PM_{7.0 \text{ and above}}$) using a pump for sampling at 28.3 l m$^{-1}$. These ambient viable impactors were precision-machined from high-quality aluminium plates and contained holes of decreasing diameter. The relevant particulates strike the stage when the ambient air cascades through the stages, whereas smaller particulates continue to flow through the stages until they are entrapped on the associated plate. The process of sampling aerosols using a 6-stage cascade impactor entails a methodological approach for gathering particles of varying sizes onto substrates in preparation for subsequent analysis. The general methodology and principle for sampling aerosols using the 6-stage cascade impactor is described in Appendix I (see Supporting Material).

Size-segregated particle sampling tools, such as Tisch Environmental Andersen Cascade Impactors, MOUDI, Dekati Impactors, low-volume samplers like the Harvard Impactor, and high-volume samplers utilizing cyclones or virtual impactors, provide distinct benefits based on research goals [33–36]. Andersen Cascade Impactors, which were employed in this study, are ideal for multi-stage particle size analysis and viable particle sampling, enabling researchers to collect bioaerosol data throughout coarse and fine particle spectra [36]. These impactors necessitate meticulous maintenance, regulation of particle rebound, and function at reduced flow rates, thereby constraining sample collection efficacy, particularly in low-concentration settings [35].

MOUDI, recognized for its high-resolution collection of ultrafine particles, reduces particle bounce and excels in detailed fine-particle analysis, but with increased operational expenses and complexity [33]. Dekati Impactors encompass a wide particle size spectrum and offer simplified maintenance, rendering them advantageous for general particle investigations; nevertheless, the main disadvantages of Dekati Impactors are particle bouncing coupled with multiple stage deposition [34]. Low-volume samplers, such as the Harvard Impactor, are economical and portable for regulatory compliance monitoring of specified particulate matter sizes, whereas high-volume samplers combined with cyclones or virtual impactors provide extensive sample collecting, suitable for chemical analysis [37]. These high-volume samplers, however, require reliable power supplies and may volatilize semi-volatile compounds due to heating, rendering them less appropriate for some organic studies.

The Tisch Environmental Andersen Cascade Impactors have distinct benefits compared to the MOUDI for the collection of particle PAHs across various aerosol sizes, especially in urban environments. The Andersen Impactors facilitate efficient and size-specific collection of PAHs across various aerosol fractions due to their extensive range of cut-off points and elevated flow rates. This is especially advantageous in variable urban settings, where PAHs may be dispersed across fine, ultrafine, and coarse particles. Furthermore, the compatibility of Andersen Impactors with quartz and Teflon filters renders them ideal for PAH analysis, facilitating extraction and minimizing volatilization loss, which is crucial for accurate PAH measurements.

Moreover, Andersen Impactors are recognized for their robustness, economic efficiency, and straightforward maintenance in field circumstances, rendering them ideal for prolonged utilization in contaminated areas. Conversely, MOUDI machines, although exceptionally accurate for ultrafine particles, necessitate increased maintenance and meticulous handling.

The Andersen design reduces temperature increase during sampling, therefore maintaining PAH integrity and averting volatilization, a concern associated with the finer stages of MOUDI. The Tisch Andersen Impactors are used for detailed, size-resolved PAH investigations in intricate urban and industrial environments.

## 2.2 Chemical analysis of PAHs

Soxhlet extraction, Accelerated Solvent Extraction (ASE), and Ultrasonication are prevalent techniques for the extraction of PAHs from aerosol particles, each presenting unique benefits. Soxhlet extraction, a conventional technique, is efficient for intricate matrices and yields high recovery rates; but, it is labor-intensive and necessitates substantial quantities of organic solvents, which can be expensive and raise environmental issues [38]. It functions at the boiling point of the solvent, hence constraining temperature regulation and potentially impacting the recovery of more volatile PAHs.

ASE is a more rapid and efficient alternative that utilizes less solvent than Soxhlet, rendering it suitable for high-throughput investigations. Its capacity to function at elevated temperatures and pressures improves extraction efficiency and permits condition regulation, while it poses a danger of thermal deterioration for some chemicals. ASE equipment is costly and necessitates maintenance, thereby restricting its accessibility. Ultrasonication is a readily available and economical method that employs moderate solvent usage and mild conditions to reduce thermal breakdown. Nonetheless, it may exhibit reduced efficiency for intricate samples, and outcomes can fluctuate based on handling, potentially affecting repeatability. The selection of the extraction process is contingent upon sample complexity, required throughput, and resource availability. Although Soxhlet extraction entails increased solvent consumption, which heightens environmental and financial concerns, it remains advantageous for comprehensive analyses owing to its simple configuration and capacity to operate without sophisticated equipment. Although ASE can diminish solvent usage, it necessitates considerable initial capital and ongoing maintenance due to its high-pressure operation, which may negate environmental and financial advantages in the long run. Ultrasonication, while cost-effective and minimizing solvent usage, frequently yields uneven results for complicated matrices and may lack the requisite consistency for high-quality data. Soxhlet extraction, despite being resource-intensive, is particularly advantageous for this study that prioritize data quality and seek to reduce extraction variability in complicated aerosol samples, while balancing cost, environmental impact, and data reliability. After sample collection, the quartz fibre filters were removed and subject to Soxhlet Extraction using dichloromethane (HPLC grade, Fisher Scientific). Before extraction, a known quantity of deuterated internal standard was added to the filters. The organic extract was nearly dried after vacuum concentration and evaporation under a stream of nitrogen. Before the fractionation procedure, the solution was changed to hexane and evaporated once more with nitrogen, resulting in a final volume of 2 mL. Flash chromatography was used to fractionate the hexane extract using silica gel and several solvents of varying polarities. Pongpiachan et al. (2013) and Tipmanee et al. (2012) provided full descriptions of the PAHs fractionation approach [39,40].

The fractionation and clean-up process adhered to the methodology outlined by Gogou et al. (1996) [41]. Following extraction, the dichloromethane (DCM) solvent was concentrated to dryness through a combination of rotary evaporation and gentle nitrogen stream blowing. The concentrated extract was then diluted in 10 mL of $n$-hexane and applied to the top of a disposable silica gel column for fractionation. The extract was fractionated into distinct compound classes using flash chromatography on silica gel. Specifically, the concentrated sample was introduced to the top of a $30 \times 0.7$ cm silica gel column containing 1.5 g of silica gel, which

had been activated at 150°C for 3 h. Nitrogen pressure was applied to achieve a flow rate of 1.4 mL min⁻¹ at the column's base. The fractionation process utilized the following eluents: (*i*) 15 mL of *n*-hexane to elute light molecular weight PAHs (Fraction 1); and (*ii*) 15 mL of a toluene-*n*-hexane mixture (5.6:9.4) to elute middle and heavy molecular weight PAHs (Fraction 2). Due to the toxicity of some solvents and the solubility properties of PAHs, toluene was selected as the preferred solvent. After fractionation, the eluates were further concentrated using a rotary evaporator followed by evaporation under a gentle nitrogen stream (flow rate set at 1.0 mbar). To address the low volatility of toluene, acetone (5–25%) was added to enhance dissipation. The concentrated sample was subsequently reduced to near-dryness and reconstituted to a final volume of exactly 100 μL using cyclohexane. This reconstituted sample was then transferred into a GC/MS vial insert for analysis.

Gas chromatography-mass spectrometry was used to identify and quantify the fractions (Shimadzu). The following 12 PAHs were observed: phenanthrene (Phe), anthracene (An), fluoranthene (Fluo), pyrene (Pyr), benz[a]anthracene (B[a]A), chrysene (Chry), benzo[b] fluoranthene (B[b]F), benzo[k]fluoranthene (B[k]F), benzo[a]pyrene (B[a]P), indeno[1,2,3-cd]pyrene (Ind), dibenz[a,h]anthracene (D[a,h]A], and benzo[g,h,i]perylene (B[g,h,i]P). Every solvent was of HPLC quality and was acquired from Fisher Scientific. The following 12 native PAHs were included in a mix standard solution (NS 9815: S-4008–100-T): Phe, An, Fluo, Pyr, B[a]A, Chry, B[b]F, B[k]F, B[a]P, Ind, D[a,h]A, and B[g,h,i]P. Two deuterated internal standards, $d_{12}$-perylene ($d_{12}$-Per) and $d_{10}$-fluorene ($d_{10}$-Fl), were acquired from Chiron AS (Stiklestadveine 1, N-7041 Trondheim, Norway). Nonane was used to prepare standard stock solutions of native and deuterated PAHs. Working solutions were obtained by diluting the mixture in cyclohexane.

A mass spectrometer is a sophisticated analytical instrument designed to separate ions based on their mass-to-charge ratio (m/z) and determine their relative abundances. The calibration of the instrument is performed using ions with known m/z values. Mass spectrometers universally operate by isolating gas-phase ions in a low-pressure environment, utilizing the influence of magnetic or electric fields on charged particles. In this study, a Shimadzu GCMS-QP2010 Ultra system was employed for analysis. This advanced instrument features a high-speed performance system with an Advanced Scan Speed Protocol (ASSP) function, enabling a maximum scan speed of 20,000 μs⁻¹ and ultra-fast data acquisition capabilities for comprehensive two-dimensional gas chromatography (GC×GC). The system's rod bias voltage is automatically optimized during high-speed data acquisition, minimizing the sensitivity loss typically associated with scan speeds exceeding 10,000 μs⁻¹. With a sensitivity five times greater than conventional systems, the GCMS-QP2010 Ultra is highly effective for applications involving fast GC/MS and comprehensive GC/MS techniques (Patent: US6610979). The separation of target compounds was achieved using a capillary column with a 60 m length × 0.25 mm internal diameter, coated with a stationary phase of 0.25 μm film thickness. The stationary phase consisted of cross-linked/surface-bonded 5% phenyl and 95% methylpolysiloxane, in accordance with specifications from various EPA methods (e.g., 207, 508, 515, 524.2, 525, 8081, 8270). A helium carrier gas (purity: 99.999%) was used at a constant flow rate of 1.0 mL min⁻¹ with a pressure pulse of 25 psi applied for 0.50 minutes. All injections (1 μL) were performed in splitless mode through a universal injector, with standards introduced using a 10 μL Hamilton syringe. The gas chromatographic oven was programmed to maintain an initial temperature of 40°C for 1 minute, followed by heating at a rate of 8°C min⁻¹ to 300°C, which was then held for 45 minutes. These chromatographic conditions enabled the qualitative resolution of several pairs of commonly co-eluting isomers, including phenanthrene/anthracene (Ph/An) and benzo[a]anthracene/chrysene (B[a]A/Chry).

## 2.3 Quantification and QA/QC

To identify the molecular markers, the quantification ions were compared to the retention times of authentic standards within a range of ± 0.2 min. Chemicals were quantified using an Internal Standard (*IS*) method (Pongpiachan et al., 2012). To be used effectively, an *IS* must have comparable physicochemical characteristics or the same kind of substitution as the analytes. A relative response factor was calculated for every native analyte. Because the relative response between the native analyte and *IS* should always be the same, this was used for quantification. This technique is convenient as the recovery losses of the compound during extraction and analysis are assumed to be equal to those of the *IS*. All QA/QC analyses were conducted using SRM1941b, which is a reference material used in many scientific and industrial applications for quality assurance and control. They were employed to guarantee the precision and dependability of the analytical data, calibrate instruments, and validate the measurement techniques. The average percentage accuracy of all PAH congeners was 89 ± 24% with the application of SRM1941b.

## 2.4 Risk assessment analysis

The total toxic equivalent quotient (*TEQ*) is a widely employed factor in toxicology and risk assessment. This study evaluated the cumulative toxic impacts of various PAHs; however, particular emphasis was on B[a]P as the *TEQ* for each PAH was calculated by multiplying its concentration in a given sample using the toxic equivalence factor (*TEF*), which quantifies its relative toxicity compared to B[a]P. This method was used because, given the significant variation in toxicity levels across different PAH congeners, it is crucial to assess the toxicity of PAHs by considering the most potent form, B[a]P. The establishment of *TEF* values for various PAHs has been undertaken by the World Health Organization (WHO) and other regulatory bodies [42–44]. These values were determined based on the carcinogenicity and other hazardous qualities associated with PAHs. The equation used to compute the cumulative B[a]P-*TEQ* is

$$TEQ = \sum_i \left[ C_i \times TEF_i \right] \tag{1}$$

where $C_i$ and $TEF_i$ are the concentrations of specific PAHs and the corresponding *TEFs*, respectively.

The *TEQ* was derived by multiplying the concentration of individual *TEF* values of each PAH, as outlined in Eq. (1). Three TEQ equations, namely Eq. (2) by Nisbet and Lagoy (1992), Eq. (3) of the US EPA (1993), and Eq. (4) by Cecinato (1997) were calculated [42, 43]. The three *TEQ* formulas use the acronyms of the PAH congeners to denote their respective concentrations.

$$TEQ_{Nisbet\,and\,Lagoy} = 0.001\left(Phe + Fluo + Pyr\right) + 0.01\left(An + B[g,h,i]P + Chry\right)$$
$$+ 0.1\left(B[a]A + B[b]F + B[k]F + Ind\right) + B[a]P + D[a,h]A \tag{2}$$

$$TEQ_{US-EPA} = 0.06\left(B[a]A\right) + 0.07\left(B[b]F + B[k]F\right) + B[a]P + 0.08\left(Ind\right) + 0.6\left(D[a,h]A\right) \tag{3}$$

$$TEQ_{Cecinato} = 0.01\left(Chry\right) + 0.1\left(B[a]A + B[b]F + B[k]F + Ind\right) + B[a]P + D[a,h]A \tag{4}$$

The sum of three and four-ring PAHs (Phe, An, Fluo, Pyr, B[a]A, and Chry) is denoted as ΣPAH(3,4) and for five and six-ring PAHs (B[b+k]F, B[e]P, B[a]P, Ind, D[a,h]A, and B[g,h,i]P) is ΣPAH(5,6). The Lifetime Lung Cancer Risk (*LLCR*) for the detected PAH concentrations in six different particle sizes was calculated using Eq. (5) [45].

$$LLCR = \sum TEQ_i \times UR_{BaP} \tag{5}$$

where $UR_{BaP}$ is the inhalation cancer unit risk ($8.7 \times 10^{-5}$ m$^3$ ng$^{-1}$).

## 2.5 Multivariate statistical techniques

This study employed Principal Component Analysis (PCA) to identify the origins of aerosols based on the chemical composition of particulate PAH congeners in six-stage configurations collected from the NAQOS. The dataset underwent preprocessing, which involved the elimination of outliers and the normalization of concentrations to guarantee equal contribution of all variables to the study [46].

PCA was subsequently employed on the standardized dataset to diminish its dimensionality and ascertain principle components (PCs) that encapsulate variable combinations elucidating the variation within the dataset [47]. Principal components with eigenvalues over 1 were preserved, and varimax rotation was employed to enhance the interpretability of the source contributions [48]. The loadings of chemical species on each principal component were analyzed to ascertain potential aerosol sources, including automotive emissions, industrial operations, biomass combustions, and transboundary pollutions [5–7,9–11].

The scores of each principal component were computed to assess the contribution of each source to the aerosol mass. The results were corroborated by supplementary methods such as Positive Matrix Factorization (PMF). This method facilitated a thorough comprehension of the principal aerosol sources in the research area, aiding efforts for air quality management and pollution reduction.

- The PMF model, created by Paatero and Tapper in 1993 and 1994, is a sophisticated multivariate statistical instrument utilized for source apportionment of aerosols in ambient air [49,50]. Source apportionment involves identifying and quantifying the contributions of diverse pollution sources, including vehicular traffic, industrial emissions, and biomass combustion, to atmospheric particle matter (PM) levels [51–53]. PMF is particularly advantageous for the analysis of intricate environmental data, as it yields robust and interpretable outcomes, essential for formulating successful air quality control policies.

- To implement the PMF model, two critical types of input data are necessary: concentration data and uncertainty estimations. In this study, the data is structured in a matrix format, where each row denotes a monitoring date and each column signifies a particular chemical species (e.g., PAH congeners). These observations document temporal and spatial fluctuations in aerosol composition, essential for recognizing the distinct characteristics of diverse pollution sources. Furthermore, uncertainty estimates, which indicate the confidence level of each measurement, are integrated into the model to address measurement mistakes [54,55].

- The PMF model then decomposes the input data matrix into two principal matrices: a source profile matrix (*f*) and a source contribution matrix (*g*). A speciation data set can be represented as a data matrix *x* with dimensions *i* by *j*, where *i* denotes the number of samples and *j* indicates the observed chemical species, accompanied by uncertainty. The objective of receptor models is to resolve the chemical mass balance (CMB) between observed

species concentrations and source profiles, as delineated in Equation 6, incorporating the number of factors *p*, the species profile *f* of each source, and the mass contribution *g* from each factor to each individual sample (refer to Eq. (6)):

$$x_{ij} = \sum_{k=1}^{p} g_{ik} f_{kj} + e_{ij} \tag{6}$$

The source profile matrix encompasses the chemical compositions or "signatures" of each pollution source, whereas the source contribution matrix delineates the extent to which each source contributes to the observed concentrations at various periods or locations. By reducing residuals and integrating uncertainty, PMF enables researchers to precisely quantify the contributions of various pollution sources without necessitating prior knowledge of source attributes, rendering it a versatile and broadly applicable instrument in air quality research. The "EPA PMF 5.0" was employed in this investigation (https://www.epa.gov/sites/default/files/2015-03/epa_pmf_5.0_setup.exe). Details on the PMF model and the method detection limit (MDL) employed in this research can be found in Supporting Materials. This study tested 3–8 factors to achieve the best solution. 100 runs were conducted for each computation to ensure that the estimated output correlates well with the detections. Since the content obtained from this study was greater than the MDL provided, the computation is based on the recommendation of EPA PMF5.0 User Guide as displayed in Eq. (7).

$$Uncertainty = \sqrt{\left(Error\,Fraction \times concentration\right)^2 + \left(0.5 \times MDL\right)^2} \tag{7}$$

The bootstrap uncertainty prediction method was employed to confirm the robustness of the statistics [49,50]. The factor mapping rates of the selected schemes all exceed 90%, suggesting that the analytical results are comparatively stable. More details on the quality control of PMF results were clearly written in Table S2–S3 (see Supporting Material).

## 3. Results and discussion

### 3.1 Meteorological conditions and atmospheric concentrations of PAHs in six different particle fractions

The meteorological data gathered at the weather station adjacent to NAQOS from January to April 2023 provides insight into the potential impact of weather conditions on the distribution and amounts of PM (see Table 1). The wind data indicates a predominant southeast direction with moderate velocities, suggesting generally tranquil conditions. Reduced wind speeds (WS) may result in the buildup of pollutants in the vicinity due to diminished dispersion. Conversely, increased wind variability, as evidenced by the standard deviation in direction (i.e., ± 68°), can intermittently facilitate the dispersion of pollutants from the monitoring location.

Table 1. Statistical description of meteorological parameters obtained from this study.

| | WD [°] | WS [m s⁻¹] | Temp [°C] | RH [%] | HI [°C] | AP [hPa] | Prec [mm] | Vis [m] |
|---|---|---|---|---|---|---|---|---|
| Aver | 130 | 1.7 | 28 | 67 | 32 | 1,010 | 0.6 | 14,753 |
| Stdev | 68 | 0.56 | 2 | 15 | 5 | 2 | 5.1 | 4,780 |
| *n* | 96 | 96 | 96 | 96 | 96 | 96 | 96 | 96 |

Note that WD, WS, Temp, RH, HI, AP, Prec, Vis, Aver, and Stdev stand for wind direction, wind speed, air temperature, relative humidity, heat index, atmospheric pressure, precipitation, and visibility, respectively.

The air temperature (AT) and relative humidity (RH) were moderate throughout this period with the average values of 28 ± 2 °C and 67 ± 15%, respectively. Elevated RH levels can result in particle development, when water vapor condenses on particulate matter, augmenting particle size. This directly affects the various PM fractions observed, as RH may determine which particles remain suspended in the air and which settle. AT indirectly influences pollution levels by affecting the activity of pollution sources, such as automobile emissions, which can fluctuate with weather conditions.

The heat index, which accounts for both AT and RH, indicates comparatively moderate perceived temperatures. Consistent atmospheric pressure readings were recorded, signifying stable weather conditions devoid of substantial fluctuations that can affect air mixing layers. This stability indicates restricted vertical dispersion of contaminants, potentially causing them to remain near the surface, particularly on days with minimal wind.

Insignificant precipitation and elevated visibility indicate primarily arid and clear weather. In the absence of precipitation, there is restricted "washout" or elimination of airborne particles, resulting in extended particle residence durations. The generally strong visibility suggests that significant particulate accumulation was not constantly problematic, although intermittent reductions in visibility may correspond with days of heightened local pollution. Collectively, these meteorological variables establish a framework for examining pollutant dynamics at NAQOS and highlight the direct influence of AT, RH, and WS on particulate matter concentrations and dispersion in the area.

The arithmetic mean concentrations of $\Sigma PAH_{12}$ collected at $PM_{0.65-1.1}$, $PM_{1.1-2.1}$, $PM_{2.1-3.3}$, $PM_{3.3-4.7}$, $PM_{4.7-7.0}$, and $PM_{7.0 \text{ and above}}$ were 13.6 ± 7.44 pg m$^{-3}$, 19.2 ± 8.37 pg m$^{-3}$, 24.3 ± 15.0 pg m$^{-3}$, 17.8 ± 7.65 pg m$^{-3}$, 13.0 ± 6.90 pg m$^{-3}$, 5.71 ± 2.41 pg m$^{-3}$, and respectively (see Table 2). This study examined fluctuations in the arithmetic mean concentrations of $\Sigma PAH_{12}$ gathered at six different PM size fractions. Further examination of the spatial distribution of the concentrations of particulate PAHs was conducted at the given locations, revealing substantial disparities in the degree of pollution, as shown in Table 3. These regions can be classified into several concentration groups based on a quantitative analysis of their spatial distribution:

**High density regions (148–271 ng m$^{-3}$).** The highest concentrations of PAHs have been observed in North China. This area includes densely populated and highly industrialised areas

Table 2. Statistical description of PAH contents (pg m$^{-3}$) in $PM_{7.0 \text{ and above}}$, $PM_{4.7-7.0}$, $PM_{3.3-4.7}$, $PM_{2.1-3.3}$, $PM_{1.1-2.1}$, and $PM_{0.65-1.1}$.

| | $PM_{7.0 \text{ and above}}$ | | $PM_{4.7-7.0}$ | | $PM_{3.3-4.7}$ | | $PM_{2.1-3.3}$ | | $PM_{1.1-2.1}$ | | $PM_{0.65-1.1}$ | |
|---|---|---|---|---|---|---|---|---|---|---|---|---|
| | Aver | Stdev | Aver | Stdev | Aver | Stdev | Aver | Stdev | Aver | Stdev | Aver | Stdev |
| Phe | 0.140 | 0.204 | 0.173 | 0.335 | 0.057 | 0.100 | 0.218 | 0.441 | 0.211 | 0.357 | 0.187 | 0.319 |
| An | 0.685 | 0.673 | 0.407 | 0.628 | 0.600 | 0.864 | 0.724 | 0.819 | 1.089 | 0.831 | 0.688 | 0.539 |
| Fluo | 0.560 | 0.616 | 1.378 | 1.985 | 1.734 | 1.564 | 2.111 | 2.505 | 2.075 | 1.917 | 1.101 | 0.858 |
| Pyr | 0.670 | 0.767 | 1.601 | 2.273 | 2.085 | 1.785 | 2.667 | 3.120 | 2.501 | 2.045 | 1.279 | 1.120 |
| B[a]A | 0.394 | 0.478 | 1.071 | 1.476 | 0.822 | 1.131 | 0.897 | 1.700 | 1.080 | 1.506 | 0.719 | 1.079 |
| Chry | 0.538 | 0.686 | 1.275 | 2.096 | 1.024 | 1.322 | 0.758 | 1.646 | 1.371 | 1.806 | 0.818 | 0.980 |
| B[b]F | 1.210 | 1.399 | 3.596 | 4.163 | 4.705 | 4.126 | 7.686 | 10.454 | 4.843 | 5.565 | 3.692 | 5.115 |
| B[k]F | 0.554 | 0.450 | 1.185 | 2.201 | 2.975 | 3.424 | 3.499 | 6.241 | 3.697 | 3.874 | 2.420 | 3.637 |
| B[a]P | 0.406 | 0.710 | 0.975 | 1.715 | 0.769 | 0.983 | 1.211 | 2.121 | 0.561 | 0.967 | 0.732 | 1.090 |
| Ind | 0.187 | 0.457 | 0.295 | 0.752 | 0.898 | 2.195 | 1.317 | 3.761 | 0.526 | 1.130 | 0.598 | 1.865 |
| D[a,h]A | N.D. | N.D. | N.D. | N.D. | 0.649 | 2.340 | N.D. | N.D. | N.D. | N.D. | N.D. | N.D. |
| B[g,h,i]P | 0.363 | 0.891 | 1.087 | 2.403 | 1.440 | 3.018 | 3.189 | 5.940 | 1.260 | 2.755 | 1.409 | 2.590 |

**Table 3. Comparison of Total PAHs contents (ng m$^{-3}$) with previous studies.**

|  | Location | Year | Number* | Particle type | Concentration | Reference |
|---|---|---|---|---|---|---|
| Total PAHs | North China | 2011 | 15 | $PM_{10}$ | 148–271 | [56] |
|  | Ulsan, South Korea | 2011 | 13 | $PM_{10}$ | 43 | [60] |
|  | Chittagong, Bangladesh | 2013 | 8 | $PM_{10}$ | 18.4 | [61] |
|  | Industrial area of Italy | 2011 | 16 | $PM_{10}$ | 5.3–8.3 | [102] |
|  | Moscow, Russia | 2002 | 12 | $PM_{10}$ | 70 | [57] |
|  | The Czech Republic | 2002 | 12 | $PM_{10}$ | 30 | [57] |
|  | Norway | 2002 | 12 | $PM_{10}$ | 0.4–1.3 | [57] |
|  | Sault Ste. Marie, Lake Superior, USA | 2011 | 21 | $PM_{10}$ | 140 | [62] |
|  | Fort McKay region, Canada | 2016 | 7 | $PM_{10}$ | 34.5 | [59] |
|  | Xi'an, China | 2015 | 16 | $PM_{10}$ | 77.7 | [58] |
|  | Shenzhen, China | 2014 | 16 | $PM_{10}$ | 3.2–81 | [63] |
|  | Shenzhen, China | 2017 | 16 | $PM_{10}$ | 30.8–69.7 | [103] |
|  | Bangkok, Thailand | 2023 | 12 | $PM_7$ | 0.0006–0.125 | This study |
|  | Xi'an, China | 2018 | 16 | $PM_{0.25}$ | 59.1 ± 54.2 | [104] |
|  | Eastern Tibetan Plateau Area | 2021 | 27 | TSP | 113 ± 189 | [105] |
|  |  |  |  | $PM_{2.5}$ | 103 ± 185 |  |
|  |  |  |  | $PM_{1.0}$ | 91 ± 177 |  |
|  |  |  |  | $PM_{0.25}$ | 46 ± 81 |  |

* Number of target pollutants.

with high automobile emissions, significant coal combustion, and air pollution [56]. High quantities are a sign of serious pollution that puts public health in danger.

**Areas of moderate concentration (30–100 ng m$^{-3}$).** Moscow ((70 ng m$^{-3}$) modest level of PAH pollution placed it within the group. Emissions from transportation and other industries may have contributed to this increase [57]. Xi'an (77.7 ng m$^{-3}$), China, was in this group, probably due to urbanisation and industrialisation [58]. The moderate concentration in the Fort McKay region of Canada (34.5 ng m$^{-3}$) can be explained by the extraction of oil sands and their associated activities [59].

**Areas of low concentration (0.4–30 ng m$^{-3}$).** Ulsan, South Korea, was within this range with a concentration of 43 ng m$^{-3}$, suggesting that pollution from industrial and traffic sources was still present, even though it was at a lower level than that in North China [60]. Chittagong (18.4 ng m$^{-3}$), Bangladesh, was in this category due to a combination of emissions from cities and industries [61]. The Czech Republic has lower PAH levels (30 ng m$^{-3}$) due to stricter environmental laws and emission controls [57]. As a result of efficient pollution management methods, Norway has some of the lowest PAH concentrations in the dataset (0.4–1.3 ng m$^{-3}$) [57]. Sault Ste. Marie, USA, has relatively high PAH levels (140 ng m$^{-3}$) and is geographically close to Lake Superior. Thus, this may be due to shipping and industrial activities [62].

**Areas of extreme variability (3.2–81 ng m$^{-3}$).** The concentrations of PAHs vary greatly within Shenzhen, China. Different pollution sources, industrial zones, and regional weather patterns are contributing factors to this fluctuation [63].

**Areas of extremely low concentration (0.0006–0.125 ng m$^{-3}$).** Our study site in Bangkok is an example of a place with very low quantities of PAHs, indicating a comparatively clean urban environment. The exceedingly low PAH content can be attributed to various factors. First, the NAQOS was located on the top roof of the Navamindradhiraj Building at an approximate height of 60 m. PAH emissions from diverse sources, such as biomass burning, industrial emissions, and vehicle releases, experience reduced dispersion and dilution at

ground level because of their proximity to emission sources. In contrast, elevated sampling positions, such as that used in this study, provide more space for dispersion and dilution, resulting in reduced PAH concentrations. The results of this study agree with those of a previous investigation conducted in Tianjin, China [64]. The PAH concentrations at a height of 40 m were higher than those at both 20 and 60 m at the urban site [64]. Second, as ultrafine particles carrying PAHs are lighter and smaller, they are prone to rapid gravitational settling at ground level. In contrast, elevated locations experience slower settling rates, allowing for greater dispersion and reduced concentrations. Third, wind patterns, turbulence, and atmospheric mixing significantly influence pollutant dispersion. Elevated structures, which are common in urban areas, can create wind patterns that enhance pollutant dispersion, further lowering PAH concentrations at relatively high sampling positions. Fourth, atmospheric PAHs can undergo chemical transformations, converting them into less toxic compounds through oxidation. Elevated temperatures at higher altitudes facilitate these reactions and decrease the PAH concentrations [65].

A distinct spatial distribution pattern was revealed by a quantitative study of the PAH concentrations throughout these regions. North China and other heavily industrialised regions typically have the highest amounts of PAHs, whereas places with more stringent environmental laws typically have lower quantities. The spatial distribution of particulate PAHs is greatly influenced by urbanisation, transportation, and industrial activity, highlighting the significance of local factors in influencing pollution levels. To manage environmental and public health issues, tracking and handling these fluctuations is essential. Overall, the data show a distinct size-dependent variation in $\Sigma PAH_{12}$ concentrations, with the highest concentrations found in fractions with finer particle sizes. This pattern is in line with earlier research showing that PAHs preferentially partition into smaller particles because of their larger surface area and improved capacity to *ad*sorb organic molecules [66]. These discoveries improve the understanding of the destiny, transit, and health effects of PAHs in the atmosphere because finer particles are easier to breathe in and can enter the respiratory system more deeply, increasing the amount of time that people are exposed to these harmful substances. To evaluate the health hazards associated with the observed changes in PAH content among different PM size fractions, further studies are required to identify the precise sources and mechanisms underlying these variations.

The quantities of PAHs in Bangkok's urban environment display unique characteristics relative to research conducted in other regions. In Chiang Mai, Thailand, PAH concentrations during the smoke haze season were approximately twice as high in rural regions than in urban areas, with fine particles (62–68%) responsible for the majority of PAHs due to biomass combustion, particularly B[b]F [67]. In Bangkok, PAHs were primarily linked to fine particles ($PM_{2.1-3.3}$, 24.3 ± 15.0 pg m$^{-3}$) and predominantly derived from automotive and industrial emissions. This discrepancy underscores the impact of emission sources and urban infrastructure, indicating that Bangkok is more influenced by traffic emissions than by biomass combustion.

A comparison with the London Underground underscores the correlation of PAHs with fine and ultrafine particles, albeit in distinct situations. In London, the average $PM_{2.5}$ concentration was 34 μg m$^{-3}$, with PAHs reaching their zenith during operational hours [68]. The rooftop location in Bangkok, albeit exhibiting lower overall concentrations, demonstrated that ultrafine particles ($PM_{0.65-1.1}$) included substantial amounts of HMW PAHs, such as B[a]P. This illustrates the ubiquitous tendency of PAHs to partition into smaller particles, capable of traveling considerable distances and infiltrating respiratory systems, highlighting the health risks associated with fine particle-bound PAHs in urban environments.

The comparison of Bangkok with indoor air investigations in Strasbourg further elucidates the parallels in PAH behavior. Indoor environments in Strasbourg exhibited overall PAH concentrations between 0.44 to 2.09 ng m$^{-3}$, with heavier PAHs (4–6 rings) predominantly linked to small particles [69]. In Bangkok, fine particulate matter (PM$_{1.1-2.1}$: 19.2 ± 8.4 pg m$^{-3}$) was predominantly composed of HMW PAHs, particularly B[b]F. Despite lower absolute concentrations in Bangkok, the distribution of PAHs in fine fractions corresponds with globally reported patterns in indoor and outdoor environments, influenced by combustion-related sources.

Comparative analyses with research from Massachusetts and other regions highlight Bangkok's unique PAH composition. In Massachusetts, PAHs were found in both fine and coarse aerosols, with LMW PAHs predominantly present in coarse particles [70]. The PAHs in Bangkok were predominantly found in fine particles, indicating that vehicular emissions are the primary source. This distinctive partitioning highlights the concentrated characteristics of PAH pollution, with Bangkok's urban landscape predominantly influenced by vehicular and industrial activity rather than biomass combustion or large-scale coal burning. These findings underscore the pivotal influence of fine particle-bound PAHs on urban air quality in Bangkok.

## 3.2 Percentage contribution of PAHs in six different particle fractions

The correlation between PAHs and the coarse fraction of airborne particles can arise from two mechanisms: particle coagulation or the volatilisation of fine particles, followed by condensation on coarse particles [71]. The morphology, chemical composition, electric charge, and hygroscopicity of particles, along with their relative humidity, are crucial elements that govern atmospheric coagulation processes [72,73]. If particle coagulation was the primary cause of PAHs' interaction with coarse particles, the concentration of PAHs (ng m$^{-3}$) in fine and coarse particles would be distributed equally. However, Fig 2 shows that this is not the case. As illustrated in Fig 2, the percentage contributions of Low Molecular Weight (LMW) PAHs, Middle Molecular Weight (MMW) PAHs, and High Molecular Weight (HMW) PAHs across six particle size fractions, ranging from PM$_{0.65-1.1}$ to PM$_{7.0 \text{ and above}}$. These results highlight the varying distribution of PAHs based on their molecular weight across different particle size ranges, providing insights into their likely sources and atmospheric behavior. The percentage contribution of HMW PAHs (5–6 ring PAHs) is dominant across all particle size fractions, consistently contributing over 50% of the total PAH composition. This trend is particularly prominent in finer particles (PM$_{0.65-1.1}$ and PM$_{1.1-2.1}$), where HMW PAHs exceed 60%. The dominance of HMW PAHs in smaller particle sizes can be attributed to their strong association with combustion sources such as diesel exhaust and other high-temperature processes. Their preferential partitioning into finer particles also suggests a potential for long-range transport and significant health impacts, as smaller particles can penetrate deeper into the respiratory system.

MMW PAHs (4-ring PAHs), represented by species such as pyrene and chrysene, contribute significantly across all particle sizes, accounting for approximately 30–40% of the total PAHs. Their higher percentages in coarser fractions (e.g., PM$_{7.0 \text{ and above}}$, PM$_{4.7-7.0}$) suggest contributions from biomass burning and mechanical re-suspension processes, which often generate coarser particles. The consistent presence of MMW PAHs across all fractions also reflects their role as intermediates between light and heavy PAHs, likely arising from mixed sources. LMW PAHs (3-ring PAHs) exhibit the lowest contributions, consistently below 10% across all particle size fractions. Their relatively higher proportion in larger particles (e.g., PM$_{7.0 \text{ and above}}$) suggests an association with volatilized emissions that have re-condensed onto coarser particles or lower-temperature processes. The lower percentages of LMW PAHs in finer fractions

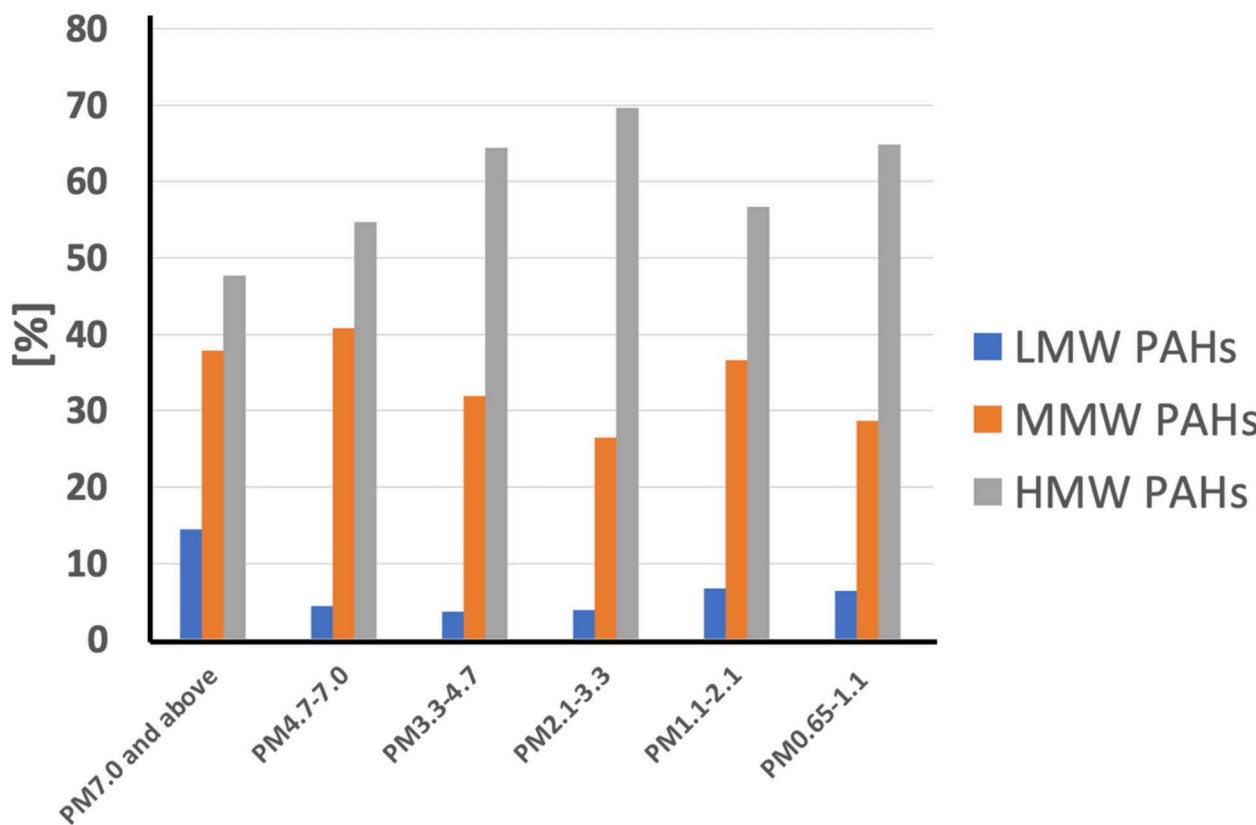

**Fig 2. Percentage contributions of ∑PAHs(3,4) and ∑PAHs(5,6) in PM$_{7.0\ and\ above}$, PM$_{4.7-7.0}$, PM$_{3.3-4.7}$, PM$_{2.1-3.3}$, PM$_{1.1-2.1}$, and PM$_{0.65-1.1}$.**

highlight their volatility and reduced stability in atmospheric conditions dominated by combustion. The distribution of PAHs across particle sizes indicates that HMW PAHs dominate in finer particles, posing a significant risk for human health due to their persistence and toxicity. MMW PAHs are evenly distributed across particle sizes, reflecting their association with multiple sources, while LMW PAHs contribute minimally and are more associated with larger particle sizes. This distribution highlights the need for targeted mitigation strategies based on particle size and associated PAH sources to reduce environmental and health impacts.

Six particle size fractions (PM$_{7.0\ and\ above}$, PM$_{4.7-7.0}$, PM$_{3.3-4.7}$, PM$_{2.1-3.3}$, PM$_{1.1-2.1}$, and PM$_{0.65-1.1}$) show different trends in particulate PAH concentration (ng g$^{-1}$) that guide both source characteristics and possible health consequences (see Fig 3). PAH congeners like An and B[b]F show the largest contributions in coarser particles (PM$_{7.0\ and\ higher}$). This pattern implies that, where these PAHs condense or adsorb, bigger particles most likely come from sources like soil dust or coarse combustion particles. Conversely, less abundant in these fractions are toxic congeners include B[a]P, Ind, and B[g,h,i]P, which indicate their inclination to combine more with ultrafine particles [74]. This finding also reflects the pattern that high molecular weight PAHs (HMW-PAHs) are greatly connected with finer particles [75].

B[b]F keeps showing dominance when particle size drops to PM$_{4.7-7.0}$ and PM$_{3.3-4.7}$; B[k]F and Pyr also start to show prominence. Given that these mid-sized particles can reach deeper lung areas than coarser particles, which would cause possible health hazards, B[a]P is particularly noteworthy in them. The steady contribution of B[b]F over these fractions highlights its connection with mid-sized combustion-related particles, probably from vehicle or industrial emissions [76,77].

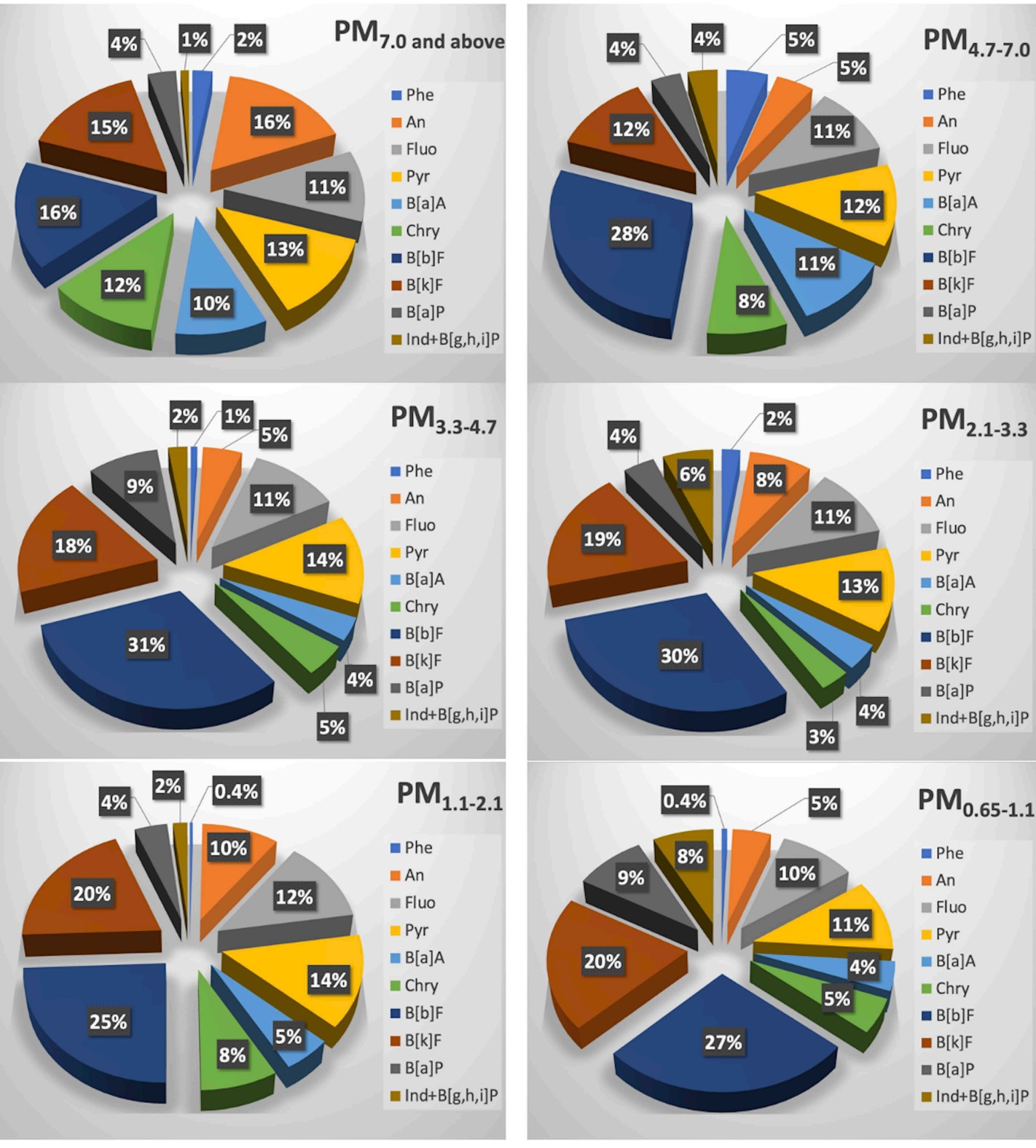

**Fig 3. Percentage contributions of particulate PAH congeners (ng g⁻¹) in PM$_{7.0\ and\ above}$, PM$_{4.7-7.0}$, PM$_{3.3-4.7}$, PM$_{2.1-3.3}$, PM$_{1.1-2.1}$, and PM$_{0.65-1.1}$.**

Whereas heavier, semi-volatile PAHs like Ind and B[g,h,i]P rise, the percentage contributions of B[k]F and An remain high in finer fractions ($PM_{2.1-3.3}$ and $PM_{1.1-2.1}$). For health effects, this size range is important as, especially those chemicals with carcinogenic qualities, particles tiny enough to reach the bronchioles are particularly hazardous due of high PAH content. The prominent presence of An in these fine fractions and Chry points to partial volatilization and re-condensation onto finer particles occurring in these compounds.

B[b]F, B[k]F, Ind and B[g,h,i]P have the largest concentrations in the finest fraction ($PM_{0.65-1.1}$). These hazardous substances have more health consequences since they can reach deeply into lung tissues. Consistent presence of B[b]F throughout all fractions, but particularly in finer particles, suggests major contributions from combustion sources, including automobile emissions. The way PAHs are distributed emphasizes overall the possible health risks connected to fine and ultrafine particles, which concentrate more hazardous and carcinogenic PAHs that the human body can inhale and absorb.

## 3.3 Diagnostic binary ratios of PAHs

The use of diagnostic binary ratios of PAHs serves as a valuable methodology for identifying potential sources of PAH contamination in the environment [9–11,21]. These ratios entail the comparison of concentrations of specific PAH components in relation to each other. Various sources of PAHs, including combustion processes, industrial operations, and natural sources, frequently provide the characteristic ratios of PAH constituents [78]. The influence of various factors, including meteorological conditions such as temperature, relative humidity, and solar radiation, combined with both heterogeneous and homogeneous reactions with trace gaseous species, can significantly affect PAH levels. Therefore, careful discretion is essential when applying diagnostic binary ratios. Notably, PAH diagnostic binary ratios exhibit significant diversity within a single source and similarity across different sources [79]. The identification of problems is particularly significant when categorising coal combustion types as it becomes challenging to differentiate between them when utilising the B[a]A/(B[a]A + Chry) ratio [80]. Nevertheless, it might be argued that these limitations are of minimal significance, as the primary source of PAHs in the ambient air of Bangkok is contamination resulting from vehicle release [9]. Furthermore, it is imperative to emphasize that coal combustion is not the main source of PAHs in Bangkok. For simplicity, it was postulated that the influence of meteorological conditions on the modification of PAH diagnostic binary ratios is minimal because of the presence of generally steady weather conditions during sample preservation.

As illustrated in Table 4, the diagnostic binary ratios of An/(An+Phe), Fl/(Fl+Pyr), and Ind/(Ind+B[g,h,i]P) detected in $PM_{7.0\,and\,above}$ obtained from this study, were in good agreement with those of $PM_{10}$ collected inside the tunnel [81], indicating the importance of traffic emissions as one of the main contributors of PAHs in the coarse particle mode. It is also interesting to note that the diagnostic binary ratios of An/(An+Phe), Fl/(Fl+Pyr), B[a]A/(B[a]A+Chry), Ind/(Ind+B[g,h,i]P), and B[a]P/(B[a]P+B[g,h,i]P) detected in $PM_{2.1-3.3}$ collected at NAQOS were 0.77, 0.44, 0.54, 0.29, and 0.28, respectively. These values differed significantly from those of $PM_{2.5}$ released from woody fuel combustion, suggesting that the impact of biomass burning as a potential source is of minor importance for ultrafine particles. Furthermore, the diagnostic binary ratios of An/(An+Phe) and Fl/(Fl+Pyr) in $PM_{2.5}$ collected from road tunnels were almost two times lower than those obtained in this study. These results indicate that a complex mixture of $PM_{2.1-3.3}$ bounded PAHs is released from various types of potential sources. As the NAQOS is adjacent to a canal boat terminal, it is reasonable to mention that the impact of canal boat emissions is inevitable.

**Table 4. Diagnostic binary ratios of PAH contents (pg m⁻³) in six different particle sizes obtained from this study in comparison with previous studies.**

| Source type | Particle size | An/(An+Phe) | Fl/(Fl+Pyr) | B[a]A/(B[a]A+Chry) | Ind/(Ind+B[g,h,i]P) | B[a]P/(B[a]P+B[g,h,i]P) | Reference |
|---|---|---|---|---|---|---|---|
| Urban ambient air | $PM_{7.0 \text{ and above}}$ | 0.83 | 0.46 | 0.42 | 0.34 | 0.53 | This study |
| | $PM_{4.7-7.0}$ | 0.70 | 0.46 | 0.46 | 0.21 | 0.47 | |
| | $PM_{3.3-4.7}$ | 0.91 | 0.45 | 0.45 | 0.38 | 0.35 | |
| | $PM_{2.1-3.3}$ | 0.77 | 0.44 | 0.54 | 0.29 | 0.28 | |
| | $PM_{1.1-2.1}$ | 0.84 | 0.45 | 0.44 | 0.29 | 0.31 | |
| | $PM_{0.65-1.1}$ | 0.79 | 0.46 | 0.47 | 0.30 | 0.34 | |
| Crop residues | $PM_{2.5}$ | 0.2 | 0.45 | 0.55 | 0.62 | 0.79 | [95] |
| Woody fuels | $PM_{2.5}$ | 0.17 | 0.44 | 0.48 | 0.56 | 0.72 | |
| Biofuels | $PM_{2.5}$ | 0.19 | 0.45 | 0.52 | 0.60 | 0.76 | |
| Crop residues | $PM_{2.5}$ | 0.13–0.45 | 0.22–0.80 | 0.33–0.50 | 0.31–0.56 | 0.23–0.67 | [106–109] |
| Woody fuels | $PM_{2.5}$ | 0.10–0.30 | 0.43–0.74 | 0.39–0.56 | 0.16–0.69 | 0.38–0.78 | [107,108,110–112] |
| Road tunnel | $PM_{10-2.5}$ | 0.014 | 0.43 | 0.53 | 0.29 | 0.31 | [113] |
| Road tunnel | $PM_{2.5-0.18}$ | 0.043 | 0.44 | 0.40 | 0.48 | 0.68 | |
| Road tunnel | $PM_{0.18}$ | 0.039 | 0.41 | 0.39 | 0.51 | 0.71 | |
| Road tunnel | $PM_{2.5}$ | 0.31 | 0.23 | 0.75 | 0.37 | 0.32 | [114] |
| Road tunnel | $PM_{10}$ | 0.40 | 0.23 | 0.75 | 0.38 | 0.32 | |
| Road tunnel | $PM_{10}$ | 0.84 | 0.46 | 0.55 | 0.34 | 0.31 | [81] |
| Road tunnel | TSP | 0.17 | 0.42 | 0.35 | 0.40 | NA | [115] |
| Road tunnel | TSP | 0.17 | 0.45 | 0.49 | 0.22 | 0.39 | [116] |

## 3.4 Pearson correlation coefficients (PCCs)

Pearson Correlation Coefficients (PCCs) are a statistical technique that can be employed to detect possible sources of PAHs in various PM size fractions. It is worth mentioning that all size fractions, including $PM_{0.65-1.1}$, $PM_{1.1-2.1}$, $PM_{2.1-3.3}$, $PM_{3.3-4.7}$, $PM_{4.7-7.0}$, and $PM_{7.0 \text{ and above}}$ were applied for PCC analysis. Correlation analysis is a valuable tool for elucidating the association between the PAH concentrations in different fractions. This analytical approach provides valuable insights into the potential sources and contributions of PAHs. As illustrated in Table 5, exceedingly strong positive PCCs were detected for Fluo *vs.* Pyr (0.982) and B[a]A *vs.* Chry (0.919). This can be attributed to several factors. A plausible explanation for the robust positive correlations observed between these pairs of PAHs is the presence of shared emission sources. Fluo and Pyr, along with B[a]A and Chry, can be emitted concurrently from similar activities or combustion processes such as traffic emissions. This interpretation is in good agreement with a previous study that reported the highest concentrations of Fluo, Pyr, and Chry in $PM_{10}$ at the entrance and exit sites of the tunnel [82].

Moderately strong positive PCCs were also observed for Ind *vs.* B[g,h,i]P, B[b]F *vs.* B[g,h,i]P, B[b]F *vs.* Pyr, and B[b]F *vs.* Fluo, with values of 0.801, 0.896, 0.892, and 0.856, respectively (Table 5). These findings are in good agreement with an earlier study indicating strong positive correlations of Ind, B[g,h,i]P, and B[b]F with other high-molecular-weight PAHs in $PM_{10}$ and $PM_{2.5}$ samples collected at a road tunnel inlet and outlet in Nanjing, China [83]. This underlines the importance of vehicular release as one of the main contributors to particulate PAHs at the study site. PAHs with comparable molecular weights frequently exhibit analogous chemical characteristics, resulting in comparable environmental behaviours. Similarities can be observed in terms of solubility, vapour pressure, and propensity to partition into particles. PAHs with comparable molecular weights can demonstrate analogous chemical stabilities

**Table 5. Pearson correlation analysis of PAH concentrations (pg m⁻³) in six different particle sizes collected at NAQOS.**

|  | Phe | An | Fluo | Pyr | B[a]A | Chry | B[b]F | B[k]F | B[a]P | Ind | D[a,h]A | B[g,h,i]P |
|---|---|---|---|---|---|---|---|---|---|---|---|---|
| Phe | 1.000 |  |  |  |  |  |  |  |  |  |  |  |
| An | 0.400 | 1.000 |  |  |  |  |  |  |  |  |  |  |
| Fluo | 0.286 | 0.457 | 1.000 |  |  |  |  |  |  |  |  |  |
| Pyr | 0.353 | 0.467 | **0.982** | 1.000 |  |  |  |  |  |  |  |  |
| B[a]A | 0.488 | 0.344 | **0.676** | **0.687** | 1.000 |  |  |  |  |  |  |  |
| Chry | 0.432 | 0.306 | **0.706** | **0.700** | **0.919** | 1.000 |  |  |  |  |  |  |
| B[b]F | 0.362 | 0.332 | **0.856** | **0.892** | **0.625** | **0.595** | 1.000 |  |  |  |  |  |
| B[k]F | 0.380 | 0.253 | **0.682** | **0.725** | **0.703** | **0.635** | **0.760** | 1.000 |  |  |  |  |
| B[a]P | **0.528** | 0.085 | 0.401 | 0.466 | **0.585** | **0.594** | 0.458 | 0.398 | 1.000 |  |  |  |
| Ind | 0.482 | 0.354 | **0.523** | **0.609** | **0.637** | **0.537** | **0.672** | **0.770** | 0.478 | 1.000 |  |  |
| D[a,h]A | −0.061 | 0.396 | 0.231 | 0.155 | 0.212 | 0.199 | −0.055 | −0.021 | −0.066 | 0.013 | 1.000 |  |
| B[g,h,i]P | 0.393 | 0.384 | **0.785** | **0.830** | **0.609** | **0.576** | **0.896** | **0.667** | 0.453 | **0.801** | −0.051 | 1.000 |

in an environmental context. This implies that these PAH congeners (i.e., Fluo, B[b]F, Pyr, B[g,h,i]P, and Ind) can remain in the environment for an extended period and engage in comparable chemical reactions, thereby strengthening their association. It is also imperative to consider the probable presence of sampling and measurement artefacts. If the sampling and analytical procedures employed for the collection and measurement of these PAHs exhibit a bias that consistently leads to their simultaneous detection, they could be responsible for significant positive associations.

### 3.5 Hierarchical cluster analysis (HCA) and principal component analysis (PCA)

Hierarchical cluster analysis (HCA) is a robust statistical methodology that can be effectively utilized to discern the potential origins of PAHs within ultrafine particles. The utilisation of HCA proved to be highly advantageous for the analysis of intricate datasets that encompass numerous factors, such as various types of PAH congeners and samples with multiple particle sizes originating from diverse sources. Previous studies have used HCA for the source identification of aerosols, terrestrial soils, and agricultural products [7,9,10,78]. In this study, an agglomerative clustering method coupled with Euclidean distance as a dissimilarity measure, in conjunction with an average linkage between groups, was applied to visualise the dendrogram, as displayed in Fig 4. The dendrogram structure depicted two main clusters of hierarchical arrangements of various PAH congeners according to their concentration patterns. Cluster I consists of B[b]F, while cluster II contains Phe, An, Fluo, Pyr, B[a]A, Chry, B[a]P, B[k]F, Ind, D[a,h]A, and B[g,h,i]P. This suggests that the PAH congeners detected in cluster II have some similarities in their concentration patterns and are not as independent as B[b]F. Prior research has established a correlation between the emissions of B[a]A, Chry, B[k]F, and B[g,h,i]P and the combustion of fuels such as diesel and petroleum. Pyr, Phe, and Fluo are tracers of the combustion of natural gas and coal [80,84,85]. Subcluster I of cluster II contained two PAH compounds: fluoranthene (Fluo) and pyrene (Pyr). The concentration profiles of these two compounds are extremely similar, which may indicate a shared origin of traffic release [86]. Subcluster II of cluster II contains Phe, An, and D[a,h]A. These compounds were categorised into this subcluster based on the similarity in their concentration profiles. Previous studies have highlighted the importance of diesel emissions as the main contributor to Phe and An [87,88]. The existence of sub-clusters within the main cluster II implies

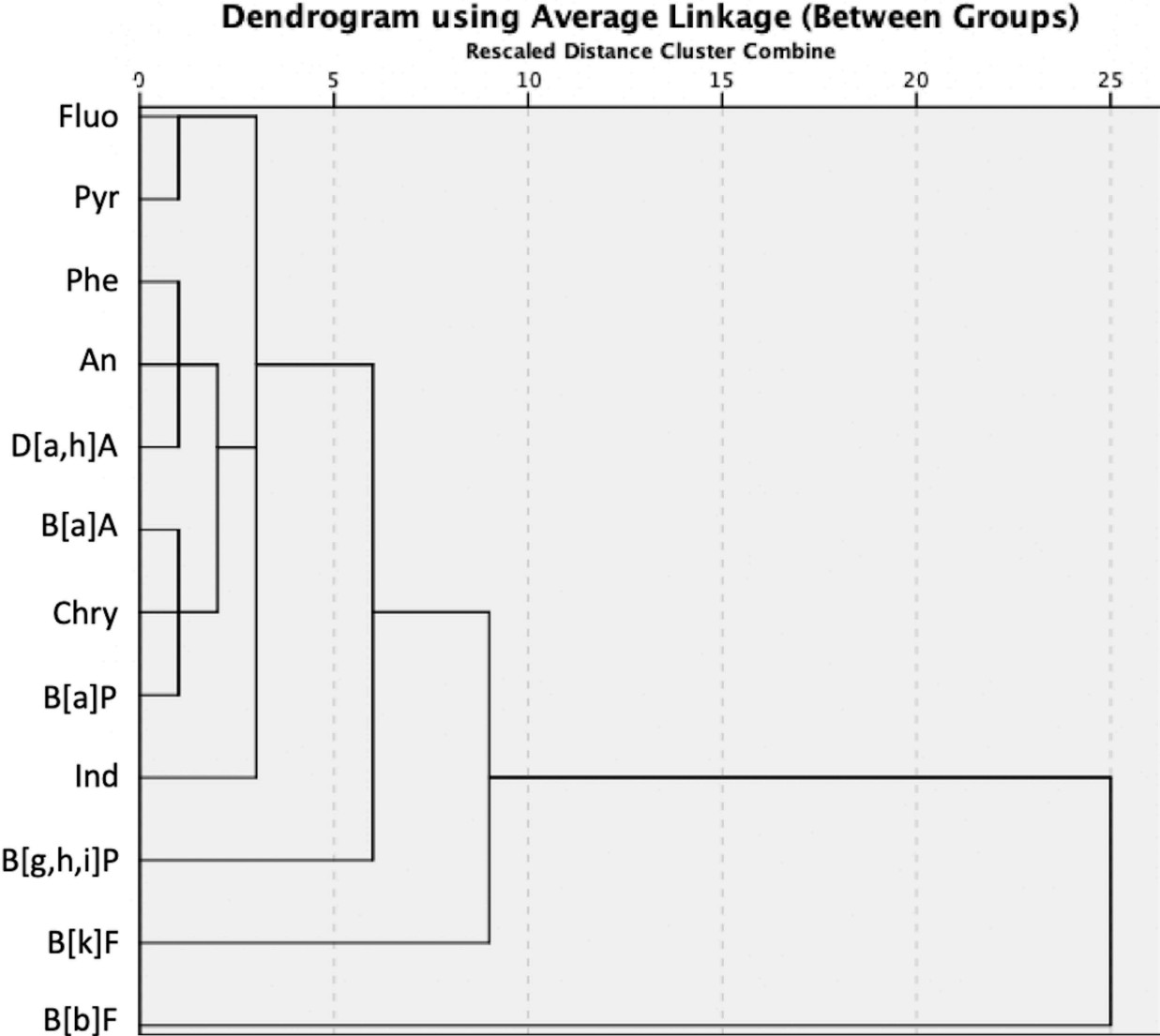

**Fig 4. Hierarchical cluster analysis (HCA) of PAHs in PM$_{7.0\text{ and above}}$, PM$_{4.7-7.0}$, PM$_{3.3-4.7}$, PM$_{2.1-3.3}$, PM$_{1.1-2.1}$, and PM$_{0.65-1.1.}$**

that these PAH compounds exhibit differing levels of similarity, potentially indicating the presence of common origins of traffic contributions. Visualising and interpreting the interrelationships among PAHs in a dataset can be facilitated using this practical tool.

PCA is an advanced statistical technique used to transform original variables into a new collection of uncorrelated variables, known as principal components (PCs). PCA was employed for the source apportionment of PAHs in ultrafine particles to ascertain and measure the contributions of various emission sources to the observed PAH concentrations. To avoid the dominance of variables with larger concentration ranges, all PAH data were normalised on the same scale. Examining the eigenvalues is crucial for assessing the significance of each PC in elucidating the variability in the dataset. PCs with higher eigenvalues can capture a larger proportion of the variation in the data [7,9,10,78].

As shown in Table 6, the varimax-rotated components of the six distinct particulate fractions were represented by principal component patterns. These patterns consisted of 12 parameters that were further divided into five PCs. PC1, PC2, PC3, PC4, and PC5 explained 57.0%, 11.9%, 9.35%, 7.39%, and 4.80% of the cumulative variance, respectively. PC1 explained significantly more variance than the other PCs. PC1 primarily consisted of Fluo, Pyr, B[b]F, and B[g,h,i]P, with relatively high correlation values of 0.90, 0.89, 0.85, and 0.77, respectively. Based on the presence of these four congeners, which are commonly associated with gasoline burning, it can be inferred that the majority (57%) of particulate PAHs originated from light-duty cars [89]. It is widely recognized that Phe, B[a]A, Chry, and B[a]P are often used biomarkers for gasohol exhausts [90]. Therefore, the elevated levels of these four PAH congeners in PC2 can be attributed to traffic emissions associated with gasohol, accounting for approximately 11.9% of gasohol emissions. Notably, the relatively high positive loadings of B[k]F and Ind observed in PC3 indicate that heavy oil combustion and electric arc furnaces in the industrial sector accounted for 10% of the emissions [91]. A study conducted by Froehner et al. (2012) found that PC4 exhibits a significant positive loading of D[a,h]A [92]. This suggests that the presence of asphalt and coal tar pavements in the construction of highways may have contributed approximately 7.4% of the overall source contribution. Given that Ph and An have been identified as tracers of diesel combustion, it is reasonable to infer that the fairly significant positive correlations observed between these two congeners in PC5 can be attributed to the contribution of canal boat diesel emissions, accounting for 4.8%, as reported by Tavares et al. (2004) [88].

**Table 6. Principal component analysis of PAH concentrations (pg m⁻³) in six different particle sizes collected at NAQOS.**

| | PC1 | PC2 | PC3 | PC4 | PC5 |
|---|---|---|---|---|---|
| Phe | 0.038 | **0.543** | 0.245 | −0.158 | **0.710** |
| An | 0.341 | −0.061 | 0.084 | 0.428 | **0.770** |
| Fluo | **0.901** | 0.262 | 0.174 | 0.225 | 0.096 |
| Pyr | **0.890** | 0.283 | 0.239 | 0.136 | 0.153 |
| B[a]A | 0.394 | **0.662** | 0.473 | 0.295 | 0.065 |
| Chry | 0.445 | **0.706** | 0.327 | 0.297 | 0.000 |
| B[b]F | **0.854** | 0.216 | 0.365 | −0.114 | 0.121 |
| B[k]F | 0.507 | 0.255 | **0.740** | 0.011 | 0.013 |
| B[a]P | 0.235 | **0.846** | 0.097 | −0.171 | 0.139 |
| Ind | 0.355 | 0.223 | **0.813** | −.045 | 0.272 |
| D[a,h]A | 0.019 | 0.007 | −0.032 | **0.936** | 0.092 |
| B[g,h,i]P | **0.766** | 0.177 | 0.450 | −0.123 | 0.230 |
| % of Total Variance | 57.0 | 11.9 | 9.35 | 7.39 | 4.80 |
| Potential Sources | light-duty gasoline vehicles | gasohol vehicles | heavy oil and electric arc furnace from industrial sector | asphalt and coal tar pavement used in the highway construction | diesel emissions from canal boats |

* Any correlation coefficients greater than 5 were highlighted in bold.

## 3.6 Positive matrix factorization (PMF)

PCA, despite its benefits, has constraints in source apportionment applications. Due to its emphasis on variation rather than actual source limitations, it might generate factors that are challenging to evaluate in practical contexts. The absence of non-negativity requirements may result negative values, which are implausible in an environmental setting. Furthermore, predominant variables can distort PCA outcomes, and associating PCA-derived factors with actual sources frequently necessitates subjective interpretation, hence constraining its accuracy and relevance for comprehensive source analysis. On the contrary, PMF offers significant benefits in managing environmental datasets by integrating measurement uncertainties, especially when data exhibit varying quality or are incomplete [49,50]. This technique also mandates non-negativity, guaranteeing that the source contributions are physically significant, a crucial aspect for realistic apportionment. Furthermore, PMF can generate adaptable source profiles without the necessity of predetermined templates, enabling it to respond to local conditions and emerging sources [93,94].

The PMF analysis of 12 PAH congeners across six particle size fractions at the NAQOS site revealed three distinct emission sources: light vehicle exhaust, biomass combustion, and diesel

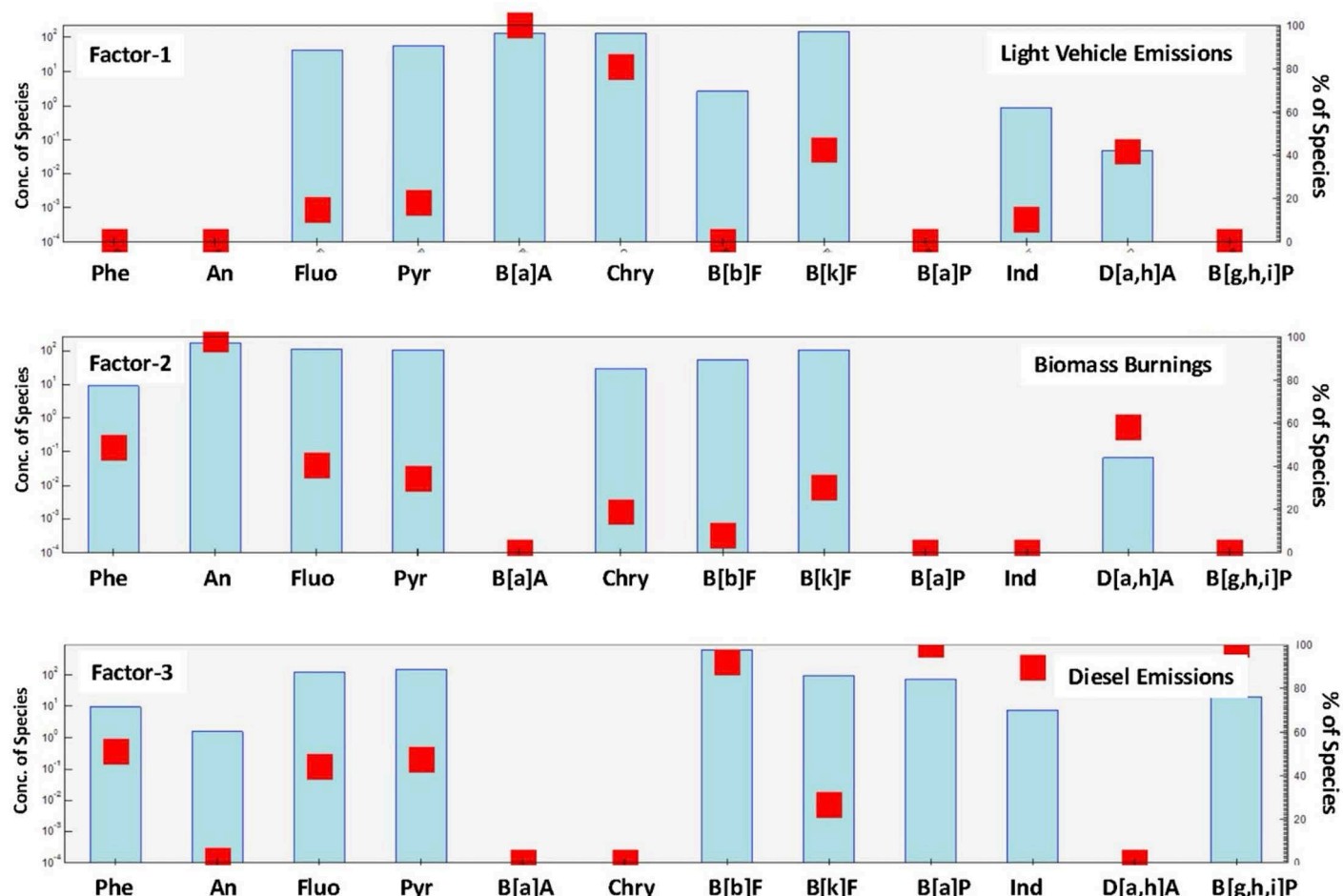

**Fig 5. Comparison of factor profile extracted by the PMF model.** A total of three factors were finally selected as the best solutions, including the light vehicle releases, biomass burnings, and diesel emissions.

emissions (see Fig 5). The primary factor (Factor 1), characterized by significant loadings of Fluo, Pyr, B[a]A, Chry, and B[k]F (all exceeding 80%), is ascribed to emissions from light vehicles, corroborated by analogous results in prior research [86,90]. This statistic indicates that PAHs are frequently emitted by gasoline-powered automobiles, possibly playing a substantial role in urban pollution from traffic sources. The second factor (Factor 2) includes D[a,h]A, Chry, B[b]F, B[k]F, Phe, An, Fluo, and Pyr, and is linked to biomass combustion, consistent with established emission profiles from open fires and agricultural waste incineration [21,95,96]. Biomass combustion significantly contributed to diverse particle sizes, especially in coarser fractions, highlighting its extensive influence on both fine and coarse particulate pollution. The third factor (Factor 3) highlights diesel emissions, characterized by elevated loadings of Ind, B[g,h,i]P, Phenanthrene (Phe), Anthracene (An), and B[a]P. These PAHs are generally released during diesel combustion and were predominantly identified in finer particles, hence corroborating the link between diesel exhaust and smaller particle fractions [77,88,97].

Further examination of the relative percentage contributions of each factor reveals valuable insights into their significance in the total PAH burden. Factor 3, accounting for 45.4% of the total PAH contributions, underscores the dominance of diesel emissions in the pollution profile at NAQOS. This is particularly evident in finer particle fractions, emphasizing the need to prioritize diesel-related sources in urban emission control strategies. Factor 1 contributes 34.2%, highlighting the substantial role of light vehicle exhaust, particularly for species like B[a]A, Chry, and B[k]F. These findings corroborate previous studies that associate urban traffic emissions with PAH contamination. In contrast, Factor 2 contributes a relatively smaller share of 20.4%, suggesting biomass combustion as a significant, yet secondary, source of PAHs. Despite its lower contribution, the influence of biomass combustion spans both fine and coarse particles, underscoring its importance in localized air quality management.

Subsequent analysis reveals that biomass combustion is a primary source of PAHs including An, D[a,h]A, Phe, Fluo, Pyr, Chry, and B[k]F across various particle sizes, underscoring its influence on both fine and coarse particles (see Fig 6). Light vehicle emissions predominantly contribute to B[a]A, Chry, B[k]F, and D[a,h]A, whereas diesel emissions are accountable for B[g,h,i]P, B[a]P, Ind, B[b]F, Phe, Fluo, Pyr, and B[k]F, particularly in finer particulate fractions. These relative percentage contributions emphasize the importance of tailoring mitigation strategies to target the most significant emission sources, particularly diesel exhaust and light vehicles, while not overlooking the notable contribution of biomass combustion to PAH levels. In summary, PMF analysis delineates the distinct contributions of light vehicles, biomass combustion, and diesel emissions to PAH profiles across particle size distributions at NAQOS. The results underscore the critical role of diesel emissions in driving urban PAH pollution, with light vehicles and biomass combustion serving as complementary contributors. These findings have significant ramifications for pollution management tactics and public health policies aimed at mitigating PAH exposure in urban settings. Integrating insights from the relative contributions of each factor enhances the specificity of source apportionment efforts, providing a robust foundation for crafting effective intervention strategies.

### 3.7 Health risk assessment of PAHs in six different particle fractions

The potential negative impacts on human health resulting from exposure to PM containing B[a]P *TEQs* can differ depending on the size of the particles and the *TEQs*. The *TEQs* for the detected PAH contents in the six different particle fractions are listed in Table 7. The *TEQs* calculated using the three different methods in descending order were $PM_{2.1-3.3}$ > $PM_{3.3-4.7}$ > $PM_{4.7-7.0}$ > $PM_{1.1-2.1}$ > $PM_{0.65-1.1}$ > $PM_{7.0\ and\ above}$. Thus, the middle fractions (i.e., $PM_{2.1-3.3}$ and $PM_{3.3-4.7}$) showed

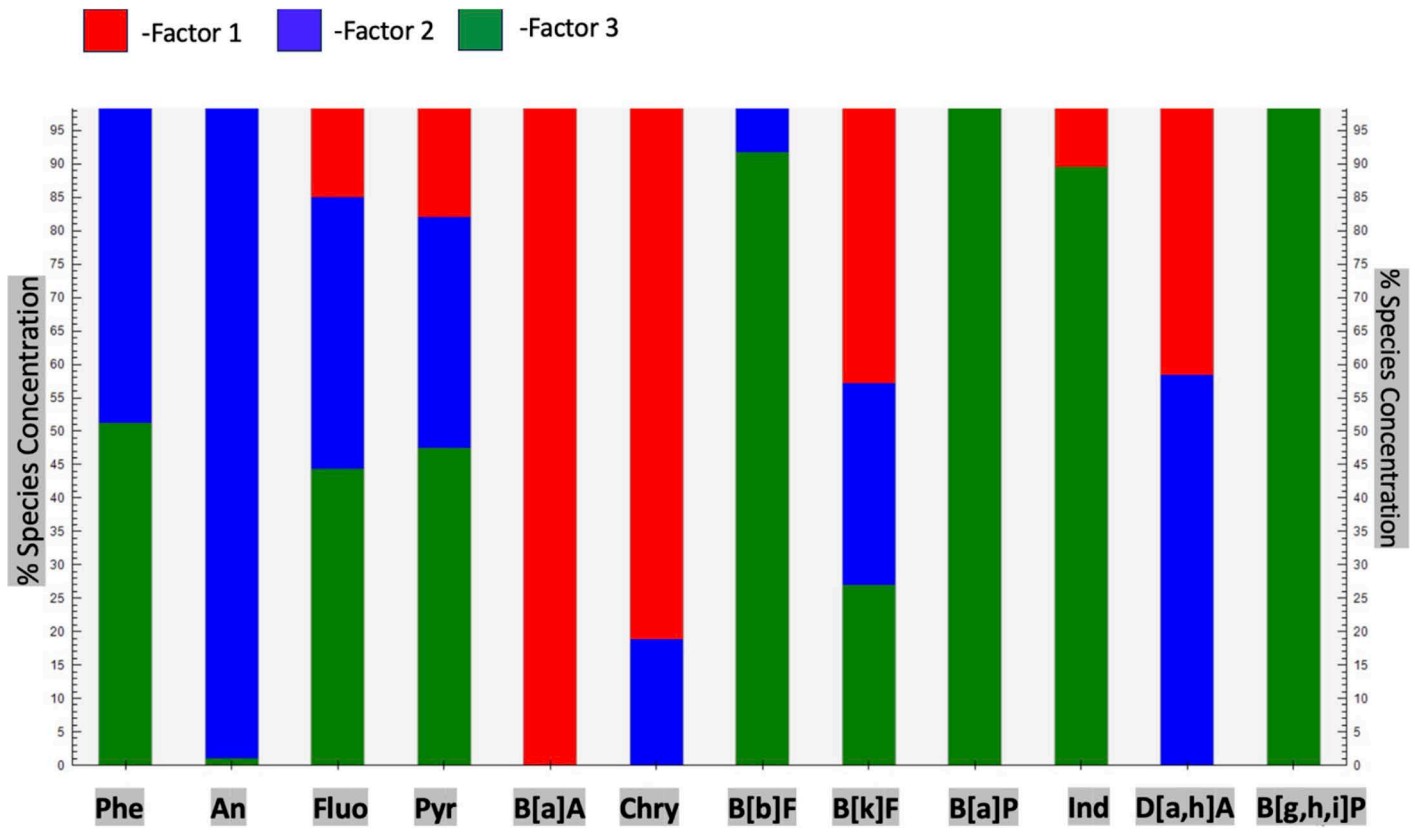

**Fig 6. Source contributions to 12 PAH congeners in PM$_{7.0 \text{ and above}}$, PM$_{4.7-7.0}$, PM$_{3.3-4.7}$, PM$_{2.1-3.3}$, PM$_{1.1-2.1}$, and PM$_{0.65-1.1}$.**

**Table 7. B[a]P toxic equivalent quotient (TEQ) for the measured PAH contents (pg m⁻³) in six different particle sizes collected at NAQOS.**

| | PM$_{7.0 \text{ and above}}$ | | PM$_{4.7-7.0}$ | | PM$_{3.3-4.7}$ | | PM$_{2.1-3.3}$ | | PM$_{1.1-2.1}$ | | PM$_{0.65-1.1}$ | |
|---|---|---|---|---|---|---|---|---|---|---|---|---|
| | Aver | Stdev | Aver | Stdev | Aver | Stdev | Aver | Stdev | Aver | Stdev | Aver | Stdev |
| TEQ$_{\text{Nisbet and Lagoy}}$ | 0.658 | 1.013 | 1.621 | 2.630 | 2.392 | 4.466 | 2.602 | 4.427 | 1.617 | 2.233 | 1.506 | 2.303 |
| TEQ$_{\text{US-EPA}}$ | 0.568 | 0.905 | 1.398 | 2.309 | 1.817 | 3.159 | 2.153 | 3.693 | 1.265 | 1.809 | 1.251 | 1.916 |
| TEQ$_{\text{Cecinato}}$ | 0.646 | 0.995 | 1.603 | 2.595 | 2.368 | 4.424 | 2.558 | 4.353 | 1.589 | 2.193 | 1.483 | 2.269 |

the greatest toxicity. In contrast, the largest and finest fractions (i.e., PM$_{7.0 \text{ and above}}$ and PM$_{0.65-1.1}$) showed the lowest toxicity with TEQ$_{\text{US-EPA}}$ values of 1.25 pg m⁻³ and 1.27 pg m⁻³, respectively.

LLCR is an essential instrument for evaluating the detrimental health consequences associated with exposure to PAHs. This analysis offers a numerical assessment of the likelihood of acquiring lung cancer as a result of exposure during an individual's lifespan. By comparing the calculated LLCRs with established benchmarks and effectively conveying the outcomes, public health workers and policymakers can make well-informed decisions aimed at safeguarding the health of both individuals and communities. In this study, all LLCR values in six different particulate fractions were below the benchmark level of 1×10⁻⁶ (see Table 8). The present study's LLCR evaluation reveals that all six particle fractions are below the threshold benchmark of 1×10⁻⁶, signifying a negligible cancer risk from PAH exposure for residents in proximity to the sampling site in Bangkok. This research offers essential insights, indicating a reduced cancer

**Table 8. Lifetime lung cancer risk (LLCR) for the detected PAH concentrations in six different particle sizes collected at NAQOS.**

| | $PM_{7.0\,and\,above}$ | | $PM_{4.7-7.0}$ | | $PM_{3.3-4.7}$ | | $PM_{2.1-3.3}$ | | $PM_{1.1-2.1}$ | | $PM_{0.65-1.1}$ | |
|---|---|---|---|---|---|---|---|---|---|---|---|---|
| | Aver | Stdev | Aver | Stdev | Aver | Stdev | Aver | Stdev | Aver | Stdev | Aver | Stdev |
| $LLCR_{Nisbet\,and\,Lagoy}$ | $5.722 \times 10^{-8}$ | $8.810 \times 10^{-8}$ | $1.410 \times 10^{-7}$ | $2.288 \times 10^{-7}$ | $2.081 \times 10^{-7}$ | $3.886 \times 10^{-7}$ | $2.264 \times 10^{-7}$ | $3.852 \times 10^{-7}$ | $1.407 \times 10^{-7}$ | $1.943 \times 10^{-7}$ | $1.311 \times 10^{-7}$ | $2.003 \times 10^{-7}$ |
| $LLCR_{US\text{-}EPA}$ | $4.942 \times 10^{-8}$ | $7.872 \times 10^{-8}$ | $1.216 \times 10^{-7}$ | $2.009 \times 10^{-7}$ | $1.581 \times 10^{-7}$ | $2.748 \times 10^{-7}$ | $1.873 \times 10^{-7}$ | $3.213 \times 10^{-7}$ | $1.101 \times 10^{-7}$ | $1.574 \times 10^{-7}$ | $1.088 \times 10^{-7}$ | $1.667 \times 10^{-7}$ |
| $LLCR_{Cecinato}$ | $5.619 \times 10^{-8}$ | $8.660 \times 10^{-8}$ | $1.394 \times 10^{-7}$ | $2.258 \times 10^{-7}$ | $2.060 \times 10^{-7}$ | $3.849 \times 10^{-7}$ | $2.225 \times 10^{-7}$ | $3.788 \times 10^{-7}$ | $1.382 \times 10^{-7}$ | $1.908 \times 10^{-7}$ | $1.290 \times 10^{-7}$ | $1.974 \times 10^{-7}$ |

risk associated with PAHs relative to other studies performed in more industrialized or extensively polluted areas. This risk assessment underscores the efficacy of LLCR as a significant instrument for public health evaluation, providing actionable data to guide policies focused on preserving air quality and reducing public health concerns.

A prior study evaluated PAH exposure in a Mediterranean industrial region and found LLCR values between $3.2 \times 10^{-5}$ and $4.3 \times 10^{-5}$, significantly surpassing the benchmark level [98]. The predominant presence of PAHs was seen in the gaseous phase, with petroleum burning and vehicular emissions identified as the principal sources. This markedly elevated risk highlights the influence of industrial pollutants, particularly from volatile PAHs, on cancer susceptibility. In contrast to NAQOS's relatively low LLCR values, the heightened levels in the Mediterranean context demonstrate how proximity to industrial activity can significantly boost cancer risk linked to PAH exposure. Another study similarly investigated PAH exposure across several urban and rural locales in the Czech Republic, revealing heightened LLCR values, especially in the industrial city of Ostrava, where risks were estimated at 1545 instances per million [99]. In rural areas, the risk of cancer persisted above the established standard. The Czech study emphasizes the impact of seasonal pollution and the role of industrial emissions in increasing PAH exposure risks, which sharply contrasts with the low cancer probability suggested by this research. This comparative investigation highlights how seasonal, localized pollution sources can exacerbate cancer risks in colder places, particularly during periods of elevated pollution.

Two prior studies offer additional insight concerning the disparities in industrial and urban exposure [100,101]. A prior study evaluated occupational PAH risks in French industries, indicating LLCR values surpassing $1 \times 10^{-4}$ in high-risk sectors such as coke production [101]. This diversity among industries further emphasizes the elevated cancer risks linked to particular occupational exposures. In contrast, a separate study conducted in Los Angeles, Milan, and Thessaloniki revealed that LLCR values in Mediterranean cities exceeded $1 \times 10^{-6}$, although Los Angeles demonstrated values below this threshold owing to stringent air quality standards [100]. These findings correspond with NAQOS's low LLCR values, highlighting the efficacy of air quality control in mitigating PAH exposure hazards in urban environments with diminished industrial contributions. These comparisons underscore the significance of emission control strategies and air quality standards in mitigating health hazards associated with PAHs.

## 3.8 Practical recommendation for policy makers

LLCR values are compared against a standard threshold of $1 \times 10^{-6}$, which represents a one-in-a-million cancer risk. Values below this benchmark are generally considered negligible in terms of health risks, while values above it indicate a higher probability of adverse health outcomes. In this study, all LLCR values for the six particle size fractions were below the $1 \times 10^{-6}$ threshold, suggesting minimal cancer risk for residents in the studied area. The findings indicate that PAH exposure at the site poses relatively low health risks compared to more industrialized or polluted urban areas. For example, studies in heavily polluted regions, such as Mediterranean industrial zones or certain Chinese cities, have reported LLCR values

significantly exceeding the threshold, highlighting the influence of local emission sources like heavy industry and coal combustion. The comparatively low LLCR values detected in this study suggest that while PAHs are present, effective regulatory measures or reduced industrial activity in Bangkok may contribute to lower risk levels. Nonetheless, finer particle fractions, which carry higher concentrations of carcinogenic PAHs, remain a concern due to their ability to penetrate deep into the respiratory system, emphasizing the importance of ongoing monitoring and mitigation strategies.

This study's findings reveal essential insights into the distribution of particle-bound PAHs and their origins, emphasizing the necessity for focused public health policies and initiatives to reduce exposure. The prevalence of diesel emissions in smaller particle fractions (45.4% of total PAHs) underscores the substantial contribution of automobile emissions to urban air pollution. These smaller particles not only contain elevated levels of carcinogenic PAHs but also possess the ability to infiltrate deeply into the respiratory system, hence increasing health risks. Policymakers should prioritize the implementation of more stringent emission rules for diesel vehicles, especially in densely populated metropolitan areas. The establishment of low-emission zones (LEZs), as effectively exemplified in cities such as London, might markedly diminish automobile contributions to PAH pollution. Promoting the adoption of electric and hybrid vehicles while systematically eliminating outdated diesel engines will significantly improve air quality.

The contributions of light vehicle exhaust (34.2%) and biomass combustion (20.4%) highlight the necessity for a comprehensive strategy for emission regulation. Public health policies ought to advocate for the adoption of cleaner fuels, such as gasohol blends with diminished PAH emission profiles, while concurrently endorsing the utilization of catalytic converters to mitigate automotive PAH emissions. Interventions for biomass combustion should target the reduction of open burning practices, especially in peri-urban regions, while promoting cleaner, energy-efficient options for cooking and heating. Public awareness campaigns may be crucial in informing communities about the health hazards associated with biomass burning and promoting safer practices.

Moreover, canal boats, although accounting for about 5% of the total PAHs, persist as a limited yet substantial source of pollution in proximity to waterways and residential zones. Regulations requiring the adoption of cleaner fuels or electric propulsion systems for canal boats might significantly diminish their emissions. Regular maintenance plans to guarantee optimal performance of boat engines and the implementation of particulate filters may also reduce their environmental impact. In addition to targeted interventions, comprehensive urban design measures, such as enhancing green infrastructure, may serve as a natural barrier to capture airborne contaminants and diminish human exposure. Investments in real-time air quality monitoring and predictive modeling could facilitate data-driven policy decisions, permitting targeted interventions in high-exposure regions. These approaches would not only reduce PAH exposure but also enhance urban air quality and public health effects.

## 4. Conclusions

This study offers essential insights into the origins, distribution, and health hazards linked to particle-bound polycyclic aromatic hydrocarbons (PAHs) in metropolitan Bangkok, concentrating on six particle size fractions. Significant findings reveal that smaller particles (e.g., $PM_{2.1-3.3}$) possess elevated levels of carcinogenic polycyclic aromatic hydrocarbons (PAHs), predominantly sourced from diesel emissions (45.4%), light vehicle exhaust (34.2%), and biomass combustion (20.4%). Despite the Lifetime Lung Cancer Risk (LLCR) values being below the threshold of $1 \times 10^{-6}$, the study highlights the potential health risks associated with ultrafine and fine particles, which can infiltrate the respiratory system and present long-term dangers, particularly to susceptible populations.

Nonetheless, many constraints in the study's design and scope must be recognized. The possible impact of unquantified sources, such household heating, cooking activities, or trans-boundary pollution, was not entirely considered, which may affect the detected PAH concentrations. The study's spatial scope was confined to a singular urban monitoring location adjacent to a canal boat dock, thus failing to encompass the heterogeneity in PAH sources and pollution patterns over the wider Bangkok metropolitan region. This research concentrated on particle-bound PAHs, excluding gaseous PAHs and other co-pollutants such heavy metals and secondary organic aerosols, which may exacerbate cumulative health concerns. Extending future study to encompass multiple monitoring locations, supplementary contaminants, and an expanded spatial range would yield a more thorough comprehension of urban air quality.

Several practical recommendations for legislators and urban planners are given based on the findings. To reduce diesel emissions, stricter vehicle emission requirements should be implemented, coupled with the encouragement of electric and hybrid vehicles in congested regions. Policies promoting cleaner fuels, updating older vehicles with sophisticated exhaust systems, and eliminating high-emission diesel engines might substantially decrease urban PAH levels. Transitioning canal boats to cleaner fuel alternatives or electric propulsion systems, together with consistent engine maintenance, would mitigate localized pollution around waterways and residential zones. Urban planners ought to incorporate green infrastructure, including vegetation barriers and urban forests, to serve as natural buffers and capture airborne contaminants in areas of high exposure. Simultaneously, public awareness initiatives ought to educate communities of the dangers of biomass combustion and advocate for sustainable alternatives. Investments in sophisticated air quality monitoring systems and predictive modeling instruments will facilitate data-driven decision-making and targeted interventions, thereby enhancing air quality, decreasing PAH exposure, and protecting human health in Bangkok and other urban areas.

## Highlights

- Higher PAH toxicities are detected in ultrafine particles.

- Traffic emissions are the main contributors of particulate PAHs.

- Canal boats account for only 5% of total PM emissions.

## Supporting information

**S1 Table. The jet orifice dimensions and particle size ranges for each stage of TE-Cascade Impactor Series 10–8XX Viable Particle Sizing Instruments.**
(DOCX)

**S2 Table. Input data statistics for PMF analysis.**
(DOCX)

**S3 Table. Base run summary table of PMF analysis.**
(DOCX)

**S1 Appendix. Cascade impactor principle, Materials and Tools, Procedure and Setup.**
(DOCX)

AckowledgementThe authors would like to express their heartfelt gratitude to the NIDA Research Centre for its invaluable support in facilitating our research. The generosity of the grant provided, coupled with access to state-of-the-art facilities, was instrumental in the successful execution of this project.

## Author contributions

**Conceptualization:** SIWATT Pongpiachan, Saran Poshyachinda.

**Data curation:** SIWATT Pongpiachan.

**Formal analysis:** SIWATT Pongpiachan.

**Funding acquisition:** Saran Poshyachinda.

**Investigation:** danai tipmanee.

**Methodology:** danai tipmanee, Phoosak Hirunyatrakul.

**Project administration:** Phoosak Hirunyatrakul.

**Resources:** Saran Poshyachinda, Phoosak Hirunyatrakul.

**Software:** Chukkapong Khumsup.

**Supervision:** Chukkapong Khumsup.

**Validation:** Chukkapong Khumsup.

**Writing – original draft:** SIWATT Pongpiachan, Muhammad Zaffar Hashmi.

**Writing – review & editing:** SIWATT Pongpiachan, Muhammad Zaffar Hashmi.

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
