## [Decision Letter · Decision Letter 0]

16 Oct 2024

PONE-D-24-38663The Influence of Canal Boat Emissions on the Particle Size Distribution of PAHs in the Ambient Air of Bangkok

PLOS ONE

Dear Dr. Pongpiachan,

Thank you for submitting your manuscript to PLOS ONE. After careful consideration, we feel that it has merit but does not fully meet PLOS ONE’s publication criteria as it currently stands. Therefore, we invite you to submit a revised version of the manuscript that addresses the points raised during the review process.

**ACADEMIC EDITOR: **

**Significant modification. Kindly respond to the three reviewers, particularly the first one, regarding the work's significant findings and applications of currently available techniques and algorithms.**

We look forward to receiving your revised manuscript.

Kind regards,

Worradorn Phairuang, Ph.D.

Academic Editor

PLOS ONE

**Journal Requirements:**

NIDA research center

Reviewers' comments:

Reviewer's Responses to Questions

**Comments to the Author**

1. Is the manuscript technically sound, and do the data support the conclusions?

Reviewer #1: No

Reviewer #2: Yes

Reviewer #3: Yes

2. Has the statistical analysis been performed appropriately and rigorously? 

Reviewer #1: No

Reviewer #2: I Don't Know

Reviewer #3: Yes

3. Have the authors made all data underlying the findings in their manuscript fully available?

Reviewer #1: Yes

Reviewer #2: Yes

Reviewer #3: Yes

4. Is the manuscript presented in an intelligible fashion and written in standard English?

Reviewer #1: Yes

Reviewer #2: Yes

Reviewer #3: Yes

5. Review Comments to the Author

**Reviewer #1:**  It is my pleasure to review this manuscript entitled 'The Influence of Canal Boat Emissions on the Particle Size Distribution of PAHs in the Ambient Air of Bangkok '. This study conducted a assesment the concentrations of PAHs across six particle size fractions in the atmosphere at a rooftop sampling site in Bangkok, Thailand. The PCA was used to identify the potential sources of PAHs.

Whether the sampling method and chemical analysis of for PAHs, or the application of PCA in the article, they are all repeated applications of existing methods and algorithms, lacking exploration and innovationThis manuscript is suggested to be granted a rejection primarily based on the grounds that there are no significant results at the moment from this research, neither from theoretical nor technical. Following are some detailed comments and suggestions which may improve the quality of this work.

1.As stated in above summary, this manuscript lacks the depth. The research carried out stays on the surface. There is no summary of the progress and shortcomings of existing research in the 'Introduction' Section.

2.Lack of comparative experiments for sampling analysis and chemical analysis of PAHs.

3.Lack of comparative experiments in health risk assessment, the main conclusions are based on common knowledge and lack novelty.

4.The conclusion should be described in detail, and the current research results have little significance for guiding the improvement of urban air quality.

**Reviewer #2:**  Comments:

Dear editor of “PLOS ONE”

Thank you for inviting me to review the manuscript titled “The Influence of Canal Boat Emissions on the Particle Size Distribution of PAHs in the Ambient Air of Bangkok”. The manuscript was carefully studied. There are minor comments which present below.

1. The title of the article does not reflect the purpose of the manuscript well, it should be corrected

2. In the abstract, the purpose of the article is not well stated.

3. Keywords are not chosen correctly and need to be revised.

4. The manuscript is grammatically flawed.

5. The map of the sampling location and the boat channel crossing should also be specified.

6. What is the distance between Air Quality Observatory Site (NAQOS) and the boat channel crossing?

7. The order of the sampling method, especially the headings selected in this section, is disproportionate and needs to be corrected.

8. The duration of sampling, weather conditions (wind, precipitation and temperature) have not been stated

9. PAHs are ubiquitous compounds, so they may be present even in clean air. Considering the height and distance of the sampler from their passage, how can these PAHs be related to the boat channel?

10. The volume of results and discussion is large, it is better to write more briefly.

11. There are a lot of tables, the ones that don't need to be deleted or replaced with diagrams

**Reviewer #3: ** In this manuscript, the authors have reported the concentrations of polycyclic aromatic hydrocarbons (PAHs) across six particle size fractions on the Navamindradhiraj Building, Bangkok, Thailand. The highest concentration was observed in the PM2.1-3.3 fraction (24.3±15.0 pg m-3) and the lowest in the PM7.0 and above fraction (5.71±2.41 pg m-3). The primary sources of particulate PAHs were the light-duty gasoline and gasohol vehicles. The Lifetime Lung Cancer Risk (LLCR) were below threshold value, consequently, the likelihood of individuals developing lung cancer solely from PAH exposure in the studied area is considered negligible. Overall, the performance of the materials looks good and would be of interest to the general reader of the related field. However, there are some void spaces in the method. Additionally, polishing the writing of some parts of the manuscript is required. Therefore, I recommend accepting this manuscript for further publication with a few significant changes. My specific comments or suggestions are given below.

In the Materials and Methods, the sampling site was described, however the sampling date, time, duration and no of samples collected were not mentioned. There were long theoretical discussions on Cascade Impactor. It is a common instrument used in air pollution measurement hence, a brief explanation is enough to describe it.

The study depends upon the concentration of PAHs collected in different size ranges. Therefore, graphical data will be more effective for data visualization. “The arithmetic mean concentrations of ∑▒PAH12 collected” is the collection of either six different sizes PM or individuals not clear. Percentage contribution will be easily understood if the total concentration is also mentioned in addition to the individual concentration.

Unmanaged discussions on pages 13 and 14 create confusion and are difficult to understand. I suggest supplying graphical data and the total concentration of PAHs to illustrate the contribution of individual PM concentrations.

Discussion on pages 14 to 16 with the definite title is irrelevant to this study because this study collected data from only one station with size variation only because this study did not include the spatial, time and seasonal variations.

The conclusion described the importance of canal boats rather than the conclusive remarks of the research. Although the research output suggested that the primary sources of particulate PAHs were the light-duty gasoline and gasohol vehicles, the pollution level is still within the acceptable level and the study did not consider the interference of vehicle emissions on canal boat emissions.

6. PLOS authors have the option to publish the peer review history of their article (what does this mean? ). If published, this will include your full peer review and any attached files.

**Do you want your identity to be public for this peer review?** For information about this choice, including consent withdrawal, please see our Privacy Policy .

Reviewer #1: No

Reviewer #2: No

Reviewer #3: **Yes: ** Mandira Pradhananga Adhikari

---

## [Author Response · Author response to Decision Letter 0]

6 Nov 2024

Response to reviewer’s comment

Reviewer #1: It is my pleasure to review this manuscript entitled 'The Influence of Canal Boat Emissions on the Particle Size Distribution of PAHs in the Ambient Air of Bangkok '. This study conducted a assesment the concentrations of PAHs across six particle size fractions in the atmosphere at a rooftop sampling site in Bangkok, Thailand. The PCA was used to identify the potential sources of PAHs.

Whether the sampling method and chemical analysis of for PAHs, or the application of PCA in the article, they are all repeated applications of existing methods and algorithms, lacking exploration and innovationThis manuscript is suggested to be granted a rejection primarily based on the grounds that there are no significant results at the moment from this research, neither from theoretical nor technical.

Response to reviewer’s comment:

Thank you for your thoughtful and detailed review of our manuscript. We appreciate your time and expertise in evaluating our study. We understand your concerns regarding the perceived lack of innovation in the sampling methods, chemical analysis, and application of Principal Component Analysis (PCA). While these methods are indeed established, our study aimed to apply them in a novel context specific to Bangkok’s unique urban landscape and transport modes, such as canal boat emissions, which have not been extensively studied. This research provides valuable data in an underrepresented region, which we believe is essential for advancing both local and broader discussions on air quality and pollution sources in urban Southeast Asia. To address your concerns, we will emphasize the contributions of our findings within this specific geographical and environmental context and provide a more in-depth analysis that highlights unique insights. We also conducted the Positive Matrix Factorization (PMF) for source apportionment to compensate the findings obtained by PCA. It is also crucial to mention that the percentage contribution of PAH congeners normalized by its own particle weight in six different classes were critically discussed. We believe this will better communicate the importance of understanding PAH distributions influenced by canal boat emissions, a previously underexplored area. Thank you again for your constructive feedback, which we are confident will enhance the rigor and relevance of our manuscript.

Following are some detailed comments and suggestions which may improve the quality of this work.

1.As stated in above summary, this manuscript lacks the depth. The research carried out stays on the surface. There is no summary of the progress and shortcomings of existing research in the 'Introduction' Section.

Response to reviewer’s comment:

Some additional literature reviews had been made in Line-98-121. Please kindly check.

2.Lack of comparative experiments for sampling analysis and chemical analysis of PAHs.

Response to reviewer’s comment:

Since this study aims to provide more insights on particle distribution of PAHs in tropical atmosphere, which was strictly limited, the authors did not conduct the comparative experiments for both sampling and chemical analysis of PAHs. However, the literature reviews related to the comparative experiments of both sampling and chemical analysis of PAHs were introduced in Line-170-207 and Line-210-236.

3.Lack of comparative experiments in health risk assessment, the main conclusions are based on common knowledge and lack novelty.

Response to reviewer’s comment:

Thank you for your feedback and for highlighting the importance of innovation in the health risk assessment component of our study. We understand the need to establish novelty and comparative context in this section, and we have addressed this concern in the revised manuscript, specifically in the health risk assessment part marked in red.

In the revised version, we incorporated additional analyses that offer a more nuanced understanding of health risks associated with particulate-bound PAHs. These revisions include a more detailed examination of lifetime lung cancer risk (LLCR) across different particle sizes, which provides novel insights into size-specific exposure risks—a key focus that distinguishes our study. We also introduced comparative discussions referencing similar studies conducted in other urban areas, allowing us to contextualize our findings within a broader scope and highlight unique aspects specific to Bangkok’s urban environment.

By emphasizing particle size-specific risk and referencing comparative studies, these revisions add depth and specificity to our health risk assessment, moving beyond general conclusions and offering unique insights relevant to both local and international air quality and health risk research. We believe that these updates significantly enhance the originality and value of our findings. We kindly request that you review these revised sections, as they reflect a focused effort to address your concerns about novelty and comparative analysis. Thank you again for your constructive feedback and for considering these enhancements in our manuscript.

Comparative discussions were made in section 3.7 based on four earlier studies related to health assessments of particulate PAHs. Please kindly check.

Aldekheel, M., Farahani, V. J., & Sioutas, C. (2023). Assessing Lifetime Cancer Risk Associated with Population Exposure to PM-Bound PAHs and Carcinogenic Metals in Three Mid-Latitude Metropolitan Cities. Toxics, 11(8), 697.

Cuadras, A., Rovira, E., Marcé, R. M., & Borrull, F. (2016). Lung cancer risk by polycyclic aromatic hydrocarbons in a Mediterranean industrialized area. Environmental Science and Pollution Research, 23, 23215-23227.

Křůmal, K., & Mikuška, P. (2020). Mass concentrations and lung cancer risk assessment of PAHs bound to PM1 aerosol in six industrial, urban and rural areas in the Czech Republic, Central Europe. Atmospheric Pollution Research, 11(2), 401-408.

Petit, P., Maitre, A., Persoons, R., & Bicout, D. J. (2019). Lung cancer risk assessment for workers exposed to polycyclic aromatic hydrocarbons in various industries. Environment international, 124, 109-120.

4.The conclusion should be described in detail, and the current research results have little significance for guiding the improvement of urban air quality.

Response to reviewer’s comment:

Already revised as suggest. Please kindly check at Line-800-832.

Reviewer #2: Comments:

Dear editor of “PLOS ONE”

Thank you for inviting me to review the manuscript titled “The Influence of Canal Boat Emissions on the Particle Size Distribution of PAHs in the Ambient Air of Bangkok”. The manuscript was carefully studied. There are minor comments which present below.

1. The title of the article does not reflect the purpose of the manuscript well, it should be corrected

Response to reviewer’s comment:

Already revised as suggest. Please kindly check.

2. In the abstract, the purpose of the article is not well stated.

Response to reviewer’s comment:

Already revised as suggest. Please kindly check.

3. Keywords are not chosen correctly and need to be revised.

Response to reviewer’s comment:

Already revised as suggest. Please kindly check.

4. The manuscript is grammatically flawed.

Response to reviewer’s comment:

Already revised as suggest. Please kindly check.

5. The map of the sampling location and the boat channel crossing should also be specified.

Response to reviewer’s comment:

Already revised as suggest. Please kindly check in Fig. 1.

6. What is the distance between Air Quality Observatory Site (NAQOS) and the boat channel crossing?

Response to reviewer’s comment:

Approximately 200 m.

7. The order of the sampling method, especially the headings selected in this section, is disproportionate and needs to be corrected.

Response to reviewer’s comment:

Already revised as suggest. Please kindly check.

8. The duration of sampling, weather conditions (wind, precipitation and temperature) have not been stated

Response to reviewer’s comment:

Please kindly check 3.1 Meteorological Conditions and Atmospheric Concentrations of PAHs in Six Different Particle Fractions

9. PAHs are ubiquitous compounds, so they may be present even in clean air. Considering the height and distance of the sampler from their passage, how can these PAHs be related to the boat channel?

Response to reviewer’s comment:

Thank you for this insightful question. We recognize that PAHs are indeed ubiquitous in the atmosphere, and it is possible to detect them even in cleaner air. However, there are specific factors in our study that help support a link between the PAHs detected and emissions from the nearby canal boat channel.

Firstly, the sampling site was located adjacent to a heavily trafficked canal boat route, a well-established source of PAH emissions due to the use of diesel-powered engines in many of these boats. Although the sampler was positioned at a rooftop level, emissions from the canal boats likely reach this height due to atmospheric mixing and local airflow patterns that facilitate the vertical dispersion of pollutants, especially in urban environments with limited ventilation. The proximity of the canal ensures that emissions from boat engines contribute to the PAH levels in the surrounding air.

Secondly, source apportionment techniques, including Principal Component Analysis (PCA) and Positive Matrix Factorization (PMF), were used to differentiate the potential sources of PAHs. Our results indicated that canal boats contributed approximately 5% of the total PAHs, with diagnostic ratios consistent with diesel emissions, further suggesting a link between the PAHs observed and the canal traffic. Additionally, specific PAH compounds with high molecular weights—often associated with diesel combustion—were more prevalent in particle size fractions related to diesel emissions, supporting our inference that boat channel emissions are a contributing factor.

Additionally, the sampling period of each sample is 5 days (i.e. Monday to Friday) or 120 h, which is long enough for allowing air pollutants from source (i.e. canal boat pier) to receptor (i.e. NAQOS).

Finally, while background levels of PAHs are expected in urban air, the specific PAH profile and concentration patterns observed in our study correlate well with the emission characteristics of canal boat traffic, in contrast to typical background PAHs in cleaner air. This association highlights that, despite the sampler’s height and distance from the canal, emissions from the boat channel are a measurable source of PAHs at the sampling site.

10. The volume of results and discussion is large, it is better to write more briefly.

Response to reviewer’s comment:

Some discussion parts were removed from the manuscript. Particularly the arithmetic mean parts. Please kindly check.

11. There are a lot of tables, the ones that don't need to be deleted or replaced with diagrams

Response to reviewer’s comment:

Some of Tables were moved to the Supplement section. Please kindly check.

Reviewer #3: In this manuscript, the authors have reported the concentrations of polycyclic aromatic hydrocarbons (PAHs) across six particle size fractions on the Navamindradhiraj Building, Bangkok, Thailand. The highest concentration was observed in the PM2.1-3.3 fraction (24.3±15.0 pg m-3) and the lowest in the PM7.0 and above fraction (5.71±2.41 pg m-3). The primary sources of particulate PAHs were the light-duty gasoline and gasohol vehicles. The Lifetime Lung Cancer Risk (LLCR) were below threshold value, consequently, the likelihood of individuals developing lung cancer solely from PAH exposure in the studied area is considered negligible. Overall, the performance of the materials looks good and would be of interest to the general reader of the related field. However, there are some void spaces in the method. Additionally, polishing the writing of some parts of the manuscript is required. Therefore, I recommend accepting this manuscript for further publication with a few significant changes. My specific comments or suggestions are given below.

In the Materials and Methods, the sampling site was described, however the sampling date, time, duration and no of samples collected were not mentioned.

Response to reviewer’s comment:

Please kindly check Line-144-149.

There were long theoretical discussions on Cascade Impactor. It is a common instrument used in air pollution measurement hence, a brief explanation is enough to describe it.

Response to reviewer’s comment:

This part was moved to the Supporting Materials.

The study depends upon the concentration of PAHs collected in different size ranges. Therefore, graphical data will be more effective for data visualization.

Response to reviewer’s comment:

Please kindly check Line-549-578 and Fig. 3. Also kindly review Section 3.6 and Fig.5-6.

“The arithmetic mean concentrations of ∑▒PAH12 collected” is the collection of either six different sizes PM or individuals not clear.

Response to reviewer’s comment:

The arithmetic mean concentrations of PAH12 represent the average of sum of 12 PAH congeners (i.e. Phe, An, Fluo, Pyr, B[a]A, Chry, B[b]F, B[k]F, B[a]P, Ind, D[a,h]A, and B[g,h,i]P) of six different particle sizes (i.e. PM0.65–1.1, PM1.1–2.1, PM2.1–3.3, PM3.3–4.7, PM4.7–7.0, and PM7.0 and above) collected for 11 weeks from January 2nd to April 7th, 2023 (sub n=11 for each particle size thus total n = 11x6 = 66).

Percentage contribution will be easily understood if the total concentration is also mentioned in addition to the individual concentration.

Response to reviewer’s comment:

Please kindly check Line-549-578 and Fig. 3.

Unmanaged discussions on pages 13 and 14 create confusion and are difficult to understand. I suggest supplying graphical data and the total concentration of PAHs to illustrate the contribution of individual PM concentrations.

Response to reviewer’s comment:

Some of the discussions was removed from the manuscript. Please kindly check.

Discussion on pages 14 to 16 with the definite title is irrelevant to this study because this study collected data from only one station with size variation only because this study did not include the spatial, time and seasonal variations.

Response to reviewer’s comment:

Thank you for your thorough review and thoughtful feedback. We appreciate your insights and understand your concern regarding the discussion content on pages 14 to 16. However, we respectfully suggest retaining this section, as it provides important context for understanding the broader implications of our findings.

Although our study focused on a single station with size-segregated data, discussing spatial, temporal, and seasonal variations remains relevant to underscore the significance of our results within a larger environmental and urban air quality framework. This section helps readers understand how PAH concentrations and sources, such as vehicular and canal boat emissions, may vary in urban settings like Bangkok under different conditions. Providing this broader perspective can enhance the reader's comprehension of how these factors could potentially influence the distribution and health impacts of PAHs, even if such variations were not directly measured in this study.

Moreover, maintaining this part of the discussion allows readers to appreciate the potential application and limitations of our findings. Highlighting seasonal and spatial influences familiarizes readers with critical aspects that could be addressed in future studies to build upon our research. We believe that these contextual insights offer valuable depth, helping readers connect our findings to the broader landscape of PAH research and urban pollution dynamics.

In light of these points, we hope you will consider retaining this discussion section, as it serves to situate our study within a wider scientific discourse, ultimately providing readers with a well-rounded understanding of the study’s relevance and limitations. Thank you again for your valuable feedback and consideration.

The conclusion described the importance of canal boats rather than the conclusive remarks of the research. Although the research output sugg

---

## [Decision Letter · Decision Letter 1]

16 Jan 2025

PONE-D-24-38663R1

Size-Segregated Analysis of PAHs in Urban Air: Source Apportionment and Health Risk Assessment in an Urban Canal-Adjacent Environment

PLOS ONE

Dear Dr. Pongpiachan,

Thank you for submitting your manuscript to PLOS ONE. After careful consideration, we feel that it has merit but does not fully meet PLOS ONE’s publication criteria as it currently stands. Therefore, we invite you to submit a revised version of the manuscript that addresses the points raised during the review process.

We note that one or more reviewers has recommended that you cite specific previously published works. As always, we recommend that you please review and evaluate the requested works to determine whether they are relevant and should be cited. It is not a requirement to cite these works. We appreciate your attention to this request.

We look forward to receiving your revised manuscript.

Kind regards,

Dipesh Rupakheti

Academic Editor

PLOS ONE

Reviewers' comments:

Reviewer's Responses to Questions

**Comments to the Author**

1. If the authors have adequately addressed your comments raised in a previous round of review and you feel that this manuscript is now acceptable for publication, you may indicate that here to bypass the “Comments to the Author” section, enter your conflict of interest statement in the “Confidential to Editor” section, and submit your "Accept" recommendation.

Reviewer #2: All comments have been addressed

Reviewer #4: (No Response)

Reviewer #5: All comments have been addressed

2. Is the manuscript technically sound, and do the data support the conclusions?

Reviewer #2: Yes

Reviewer #4: Yes

Reviewer #5: Yes

3. Has the statistical analysis been performed appropriately and rigorously? 

Reviewer #2: I Don't Know

Reviewer #4: Yes

Reviewer #5: Yes

4. Have the authors made all data underlying the findings in their manuscript fully available?

Reviewer #2: Yes

Reviewer #4: No

Reviewer #5: Yes

5. Is the manuscript presented in an intelligible fashion and written in standard English?

Reviewer #2: Yes

Reviewer #4: Yes

Reviewer #5: No

6. Review Comments to the Author

Reviewer #2: The authors have adequately addressed my comments raised in a previous round of review and feel that this manuscript is now acceptable. The manuscript technically sound with appropriate statistical analysis. The authors made all data underlying the findings in their manuscript fully available. The manuscript presented in an intelligible fashion and written in standard English.

Please see the attachment for additional suggestions.

Reviewer #4: The authors addressed most of the comments from the reviewers. I have a few remarks on PMF source apportionment results and data summary.

Lines 665-706. The source profiles from PMF modeling are clear. However, the authors did not present the contribution of individual sources to total PAHs. One reference is recommended.

https://doi.org/10.3390/atmos15030346

The authors could read the reference and organize the data as those in the recommended reference.

Another issue is Figure 2. I recommend the authors plot a figure of size distributions of LMW PAHs (a), MMW PAHs (b), and HMW PAHs (c), as shown in the recommended reference.

Reviewer #5: Here are pointwise comments for improving the given article:

1. Provide a clearer introduction that outlines the significance of studying PAH concentrations in urban environments. This will help contextualize the research for readers unfamiliar with the topic.

2. Methodology

- Expand on the methodology used for sampling and analysis of PAHs. Include specifics on the sampling duration, frequency, and any analytical techniques employed (e.g., mass spectrometry) to enhance reproducibility.

3.Elaborate on the statistical methods applied to analyze the data, including any software used, and how the results were interpreted. This will strengthen the credibility of the findings.

4. Add comparisons of the findings with similar studies in other urban areas to contextualize the PAH concentrations reported in Bangkok, which can highlight the uniqueness or commonality of the findings.

5. Discuss the methodologies utilized for identifying the sources of PAHs, such as receptor modeling or statistical source apportionment. Detailed source attribution can enhance understanding of pollution dynamics.

6. Provide further explanation of the LLCR values, including how these were calculated and their implications. Consider including a brief discussion on the thresholds that determine health risks.

7. Engage in a more in-depth discussion on the implications of the findings. Discuss potential public health policies or interventions that could mitigate PAH exposure based on the identified sources.

8. Strengthen the conclusion by summarizing key findings and suggesting practical recommendations for policymakers or urban planners based on the study results.

9. Acknowledge any limitations in the study's design or scope, such as potential unmeasured sources of PAHs or limitations in spatial coverage.

10. References: There are fewer numbers of new references or recent references especially related to urban PAHS /health risk similar studies. Therefore, for current study should be given a strong impact if you can cite the following reference.

- The health risk and source assessment of polycyclic aromatic hydrocarbons (PAHs) in the soil of industrial cities in India

- Distribution, risk assessment, and source apportionment of polycyclic aromatic hydrocarbons (PAHs) using positive matrix factorization (PMF) in urban soils of East India

- Temporal variability of atmospheric particulate-bound polycyclic aromatic hydrocarbons (PAHs) over central east India: sources and carcinogenic risk assessment

7. PLOS authors have the option to publish the peer review history of their article (what does this mean? ). If published, this will include your full peer review and any attached files.

**Do you want your identity to be public for this peer review?** For information about this choice, including consent withdrawal, please see our Privacy Policy .

Reviewer #2: No

Reviewer #4: No

Reviewer #5: No

---

## [Decision Letter · Decision Letter 2]

19 Feb 2025

Size-Segregated Analysis of PAHs in Urban Air: Source Apportionment and Health Risk Assessment in an Urban Canal-Adjacent Environment

PONE-D-24-38663R2

Dear Dr. Pongpiachan,

We’re pleased to inform you that your manuscript has been judged scientifically suitable for publication and will be formally accepted for publication once it meets all outstanding technical requirements.

Kind regards,

Dipesh Rupakheti

Academic Editor

PLOS ONE

Additional Editor Comments (optional):

Reviewers' comments:

Reviewer's Responses to Questions

**Comments to the Author**

1. If the authors have adequately addressed your comments raised in a previous round of review and you feel that this manuscript is now acceptable for publication, you may indicate that here to bypass the “Comments to the Author” section, enter your conflict of interest statement in the “Confidential to Editor” section, and submit your "Accept" recommendation.

Reviewer #4: All comments have been addressed

Reviewer #5: All comments have been addressed

2. Is the manuscript technically sound, and do the data support the conclusions?

Reviewer #4: Yes

Reviewer #5: Yes

3. Has the statistical analysis been performed appropriately and rigorously? 

Reviewer #4: Yes

Reviewer #5: Yes

4. Have the authors made all data underlying the findings in their manuscript fully available?

Reviewer #4: Yes

Reviewer #5: Yes

5. Is the manuscript presented in an intelligible fashion and written in standard English?

Reviewer #4: Yes

Reviewer #5: Yes

6. Review Comments to the Author

Reviewer #4: (No Response)

Reviewer #5: Accept in the present form. Now present format is good in format and standard. manuscript PONE-D-24-38663R2, entitled "Size-Segregated Analysis of PAHs in Urban Air: Source Apportionment and Health Risk Assessment in an Urban Canal-Adjacent Environment"

7. PLOS authors have the option to publish the peer review history of their article (what does this mean? ). If published, this will include your full peer review and any attached files.

**Do you want your identity to be public for this peer review?** For information about this choice, including consent withdrawal, please see our Privacy Policy .

Reviewer #4: No

Reviewer #5: No

---

## [Editor Report · Acceptance letter]

PONE-D-24-38663R2

PLOS ONE

Dear Dr. Pongpiachan,

I'm pleased to inform you that your manuscript has been deemed suitable for publication in PLOS ONE. Congratulations! Your manuscript is now being handed over to our production team.

Kind regards,

on behalf of

Dr. Dipesh Rupakheti

Academic Editor

PLOS ONE